# MUSS: Multilevel Subset Selection for Relevance and Diversity

## Abstract

The problem of relevant and diverse subset selection has a wide range of applications, including recommender systems and retrieval-augmented generation (RAG). For example, in recommender systems, one is interested in selecting relevant items, while providing a diversified recommendation. Constrained subset selection problem is NP-hard, and popular approaches such as Maximum Marginal Relevance (MMR) are based on greedy selection. Many real-world applications involve large data, but the original MMR work did not consider distributed selection. This limitation was later addressed by a method called DGDS which allows for a distributed setting using random data partitioning. Here, we exploit structure in the data to further improve both scalability and performance on the target application. We propose MUSS, an efficient method that uses a multilevel approach to relevant and diverse selection. In a recommender system application, our method can not only improve the performance up to 4 percent points in precision, but is also 20 to 80 times faster. Our method is also capable of outperforming baselines on RAG-based question answering accuracy. We present a novel theoretical approach for analyzing this type of problems, and show that our method achieves a constant factor approximation of the optimal objective. Moreover, our analysis also results in a $\times 2$ tighter bound for DGDS compared to previously known bound.

## 1 Introduction

**Relevant and diverse subset selection** plays a crucial role in a number of machine learning (ML) applications. In such applications, relevance ensures that the selected items are closely aligned with task-specific objectives. E.g., in recommender systems these can be items likely to be clicked on, and in retrieval-augmented generation (RAG) these can be sentences that are likely to contain an answer. On the other hand, diversity addresses the issue of redundancy by promoting the inclusion of varied and complementary elements, which is essential for capturing a broader spectrum of information. Together, relevance and diversity are vital in applications like feature selection (Qin et al., 2012), document summarization (Fabbri et al., 2021), neural architecture search (Nguyen et al., 2021; Schneider et al., 2022), deep reinforcement learning (Parker-Holder et al., 2020; Wu et al., 2023b), and recommender systems (Clarke et al., 2008; Coppolillo et al., 2024; Carraro & Bridge, 2024). Instead of item relevance, one can also consider item quality. Thus sometimes, we will refer to the problem as high quality and diverse selection.

**Challenges** of relevant and diverse selection arise due to combinatorial nature of subset selection and the inherent trade-off in balancing these two objectives. Enumerating all possible subsets is impractical even for moderately sized datasets due to exponential number of possible combinations (He et al., 2012; Gong et al., 2019; Maharana et al., 2023; Acharya et al., 2024). In addition, the combined objective of maximizing relevance and diversity is often non-monotonic, further complicating optimization. For instance, the addition of a highly relevant item might significantly reduce diversity gains. In fact, common formulations of relevant and diverse selection lead to an NP-hard problem (Ghadiri & Schmidt, 2019).

**Existing approaches** consider different approximate selection techniques, including clustering, reinforcement learning, determinantal point process, and maximum marginal relevance (MMR). Among these MMR has become a widely used framework for balancing relevance and diversity (Guo & Sanner, 2010; Xia et al., 2015; Luan et al., 2018;

Hirata et al., 2022; Wu et al., 2023a). This greedy algorithm iteratively selects the next item that maximizes gain in weighted combination of the two terms. The diversity is measured with (dis-)similarity between the new and previously selected items.

MMR algorithm is interpretable and easy to implement. However, the original MMR work did not consider distributed selection, while many real-world ML applications deal with large-scale data. This limitation was later addressed by a method called DGDS (Ghadiri & Schmidt, 2019). The authors of DGDS also provided theoretical analysis showing that their method achieves a constant factor approximation of the optimal solution. DGDS allows for a distributed setting using random data partitioning. Items are then independently selected from each partition, which can be performed in parallel. Subsets selected from the partitions are then combined before the final selection is performed. Thus, the final selection step becomes a performance bottleneck if the number of partitions and the number of selected items in each partition are large. We refer to Appendix Section A for further discussion on related work and summarize the computational complexity in Table 1.

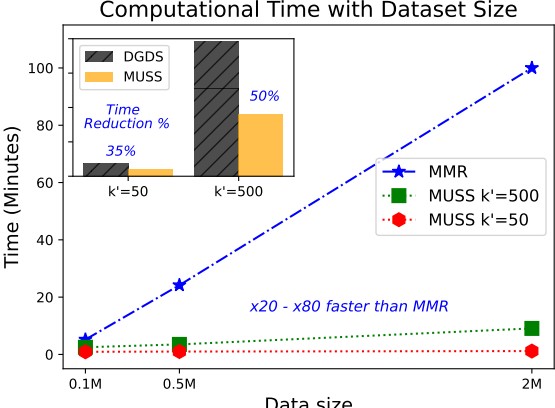

Figure 1: Our MUSS is not only capable of achieving better performance on the target task as baselines, but also can be $20\times$ to $80\times$ faster. The insert shows the relative speed improvement against DGDS. Note that MMR is not a distributed method. Here, the task has been to select $k$ candidate items for recommendation from catalogs of different sizes and $k'$ denotes the number of intermediate items to be selected within each cluster for MUSS and DGDS.

**In our work**, we explore the question *whether we can further improve both scalability and performance on the target application by leveraging structure in the data*. We address the final selection bottleneck by introducing clustering-based data pruning. Moreover, our novel theoretical analysis, such as Lemmas 1 and 5, allowed us to relate the cluster-level and item-level selection stages and derive an approximation bound for the proposed method. In summary, the contributions of our work are as follows

- We propose MUSS, an efficient distributed method that uses a multilevel approach to high quality and diverse subset selection.
- We provide a rigorous theoretical analysis and show that our method achieves a constant-factor approximation of the optimal objective. We show how this bound can be affected by clustering structure in the data.
- We utilize our new theoretical findings to tighten the bound of DGDS, improving from the existing factor of $\frac{1}{31}$ to $\frac{1}{16}$. Moreover, the improved bound does not rely on the condition of $k \geq 10$ required in DGDS (Ghadiri & Schmidt, 2019).
- We demonstrate the utility of our method on popular ML applications of item recommendation and RAG-based question answering. For item recommendation, our method not only improves up to $4$ percent points in precision upon baselines, but is also 20 to 80 times faster (Figure 1). MUSS has been deployed **in production** for real-world candidate retrieval on a large-scale e-commerce platform serving million customers daily.

## 2 MUSS: MULTILEVEL SUBSET SELECTION

### 2.1 PROBLEM FORMULATION

Consider a universe of objects represented as set $\mathcal{U}$ of size $|\mathcal{U}| = n$. Let $q : \mathcal{U} \to \mathbb{R}^+$ denote a non-negative function representing either quality of an object, relevance of the object, or a combination of both. Next, consider a distance function $d : \mathcal{U} \times \mathcal{U} \to \mathbb{R}^+$. Here we implicitly assume that the objects can be represented with embeddings in a metric space. Appendix Table 5 summarizes our notation.

Our goal is to select a subset $\mathcal{S} \subseteq \mathcal{U}$ of size $|\mathcal{S}| = k \leq n$ from the universe $\mathcal{U}$, such that the objects are both of high quality and diverse. In particular, we consider the following optimization problem

$$\mathcal{O} = \arg \max_{\mathcal{S} \subseteq \mathcal{U}, |\mathcal{S}|=k} F(\mathcal{S} \mid k, \lambda) \qquad (1)$$

where $\mathcal{O}$ is the global optimum, and the objective function is defined as

$$F(\mathcal{S}) = \lambda \sum_{\boldsymbol{u} \in \mathcal{S}} q(\boldsymbol{u}) + (1 - \lambda) \sum_{\boldsymbol{u}, \boldsymbol{v} \in \mathcal{S}} d(\boldsymbol{u}, \boldsymbol{v})$$
$$= \lambda Q(\mathcal{S}) + (1 - \lambda) D(\mathcal{S}). \qquad (2)$$

The first term $Q$ measures the quality of selection, while the second term $D$ measures the diversity of the selection. Coefficient $0 \geq \lambda \geq 1$ controls the trade-off between quality (or relevance) and diversity. A higher value of $\lambda$ increases the emphasis on quality, while a lower value emphasizes diversity thus reducing redundancy. We use $\mathcal{O}$ to denote the global maximizer of the above problem parameterized by $k$ and $\lambda$. For brevity, we may omit $k$ and $\lambda$ throughout the paper and write $F(\mathcal{S})$.

Table 1: MUSS reduces time complexity by only considering a subset of clusters. Here, we compare average-case time complexity of methods for selecting subsets of size $k$ from a set of $n$ items. We use $l$ to denote the number of clusters, $m$ for the number of selected clusters, $k'$ for the number of items to be selected in each cluster and $p$ for the number of parallel cores. Typically $n \gg l \gg m$; $l$ for DGDS does not have to be the same as $l$ for MUSS. Complexity of MUSS is discussed in Section 2.2. For DGDS and MUSS partitioning and clustering steps can be performed once and are not considered here.

| Method | Computational Complexity |
|--------|--------------------------|
| K-DPP | $\mathcal{O}(k^2 n + k^3)$ |
| MMR | $\mathcal{O}(k^2 n)$ |
| DGDS | $\mathcal{O}\left( \frac{(k')^2 n}{p} + k^2(k'l) \right)$ |
| MUSS | $\mathcal{O}\left( m^2 l + \frac{(k')^2 nm}{lp} + k^2(k'm + k) \right)$ |

Note that the entire objective can be multiplied by a positive constant without changing the optimal solution. As such, different scaled variations of the diversity term can be represented with the same objective. For example, one can consider using an average distance for diversity, and this would lead to the same optimization problem with a different choice of $\lambda$.

The optimization involves maximizing a function with a cardinality constraint, which is a well-known NP-hard problem. Therefore, our solution uses a greedy selection strategy similar to MMR. However, a direct application of MMR might not be practical for large sets. Distributed approach of DGDS partially addresses this problem, but it still has a bottleneck in the final selection from the union of points selected from partitions.

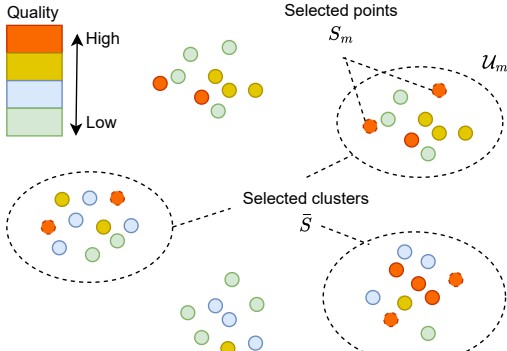

Figure 2: MUSS performs clustering following by a multilevel selection. Here, $\bar{\mathcal{S}}$ is a set of selected clusters, $\mathcal{U}_m$ denotes cluster $m$, and $\mathcal{S}_m$ denotes items selected from that cluster.

## 2.2 MULTILEVEL SELECTION

We address this bottleneck by considerably reducing the size of this union without compromising quality of selection. To this end, we propose MUSS, a method that performs selection in three stages: (i) selecting clusters, (ii) selecting objects within each selected cluster, and (iii) selecting the final set from the union of objects selected from the clusters (Figure 2). We show that MUSS achieves a constant factor approximation of the optimal solution.

**Step 1:** While in previous literature greedy selection has been applied to items, our key observation is that greedy selection can also be used to select entire clusters that are both diverse and of high-quality while filtering out other clusters thus reducing the total pool of candidate items.

Therefore, we can use KMEANS algorithm to partition the data into clusters $\mathcal{U} = \bigcup_{i=1}^{l} \mathcal{U}_i$. Other clustering algorithms could also be used at this step. Next, we view clusters as

a set of items $\mathcal{C} = \{c_1, \ldots, c_l\}$. The distance $d(c_i, c_j)$ between two clusters is defined as the distance between cluster centroids. Next, the quality of the cluster is defined as the median quality score of items in this cluster, i.e., $q(c_i) = \text{median}(\{q(a) : a \in \mathcal{U}_i\})$. We then apply Algorithm 1 with the set of clusters $\mathcal{C}$ as input.

**Step 2:** Using greedy selection at the cluster level will result in a subset of $m$ selected clusters, where each cluster $c_i$ contains items $\mathcal{U}_i \subset \mathcal{U}$. For each selected cluster, we independently apply Algorithm 1 to select $\mathcal{S}_i = \text{ALG1}(\mathcal{U}_i | k')$ where $|\mathcal{S}_i| = k'$. We can set $k' < k$ for computational speed up (see Fig. 1). Importantly, selections within different clusters can be executed in parallel.

---

**Algorithm 1** Greedy Selection

**Input:** set $\mathcal{T}$, #items to select $k$, parameter $\lambda \in [0, 1]$
**Output:** set $\mathcal{S} \subseteq \mathcal{T}$, s.t. $|\mathcal{S}| = k$
    `// start with the highest quality item`
1  $\mathcal{S} = \{\arg\max_{t \in \mathcal{T}} q(t)\}$
2  **for** $i = 2, \ldots, k$ **do**
3     $s = \arg\max_{t \in \mathcal{T} \setminus \mathcal{S}} \lambda q(t) + (1 - \lambda) \sum_{u \in \mathcal{S}} d(t, u)$
4     $\mathcal{S} = \mathcal{S} \cup \{s\}$

---

**Step 3:** Different from DGDS, our final selection includes the top $k$ items with the highest overall quality.[1] That is, we collect $\mathcal{S}^* = \arg\max_{A \subseteq \mathcal{U}, |A| = k} \sum_{u \in A} q(u)$. We then select the final set of items by applying Algorithm 1 on the union of item sets obtained in the previous step combined with $\mathcal{S}^*$. Our final selection is $\mathcal{S} = \text{ALG1}(\cup_{i=1}^m \mathcal{S}_i \cup \mathcal{S}^* | k)$ where $|\mathcal{S}| = k$. The entire approach is summarized in Algorithm 2.

**Computational complexity:** We discuss the average-case time complexity of MUSS. Here "average-case" means assuming cluster sizes of $\frac{n}{l}$ when clustering $n$ items into $l$ clusters. The complexity of standard iterative implementation of KMEANS algorithm (Lloyd, 1982) is $\mathcal{O}(nkt)$, where $t$ is the number of iterations. Selecting the top $k$ highest quality items $\mathcal{S}^*$ is precomputed in the candidate retrieval task. In cases of computing them from scratch with a distributed setting, it costs $\mathcal{O}(n + pk \log k)$ using min-heap where $p$ is the number of parallel cores. Greedy selection of $k$ out of $n$ items can be performed in $\mathcal{O}(k^2 n)$ time. Therefore, selecting $m$ out of $l$ clusters results in $\mathcal{O}(m^2 l)$. Next, selection of $k' \le k$ points within one cluster gives $\mathcal{O}(\frac{(k')^2 n}{l})$. This will only be performed for $m$ selected clusters and the computation can be dis-

---

**Algorithm 2** MUSS

**Input:** set $\mathcal{U}$; item-level parameters: #items to select within each cluster $k_w$ and globally $k$, trade-off $\lambda$; cluster-level parameters: #clusters $l$, #clusters to select $m$, trade-off $\lambda_c$
**Output:** $\mathcal{S} \subseteq \mathcal{U}$ with $|\mathcal{S}| = k$
1  Apply KMEANS$(\mathcal{U}, l)$ to cluster $\mathcal{U}$ into $\{\mathcal{U}_i\}_{i=1}^l$
2  Let $\mathcal{C}$ denote a set of clusters. The distance and quality of clusters are defined in Section 2.2.
3  $\bar{\mathcal{S}} = \text{ALG1}(\mathcal{C} | m, \lambda_c)$
4  **for** $\mathcal{U}_i \in \bar{\mathcal{S}}$ **do**
     `// selection within in each cluster`
5     $\mathcal{S}_i = \text{ALG1}(\mathcal{U}_i | k', \lambda)$
    `// the top k highest quality items`
6  $\mathcal{S}^* = \arg\max_{A \subseteq \mathcal{U}, |A| = k} \sum_{u \in A} q(u)$
    `// refinement for final selection`
7  $\mathcal{S} = \text{ALG1}(\cup_{i=1}^m \mathcal{S}_i \cup \mathcal{S}^* | k, \lambda)$

---

tributed across $p$ cores resulting in $\mathcal{O}(\frac{(k')^2 nm}{lp})$. Combining subsets from the clusters and the top $k$ highest quality items results in a pool of $k'm + k$ items. Thus the final selection step results in $\mathcal{O}(k^2(k'm + k))$ complexity. Clustering and global top-k quality selection are performed once. Thus at query time, average-case complexity is $\mathcal{O}\left(m^2 l + \frac{(k')^2 nm}{lp} + k^2(k'm + k)\right)$. Since our approach does not train a separate model for data selection, it does not require extra space. Therefore, the memory complexity is linear in the data size.

## 2.3 THEORETICAL PROPERTIES

We now present theoretical analysis of the proposed algorithm. We show that MUSS achieves a constant factor approximation of the optimal solution. Our main results are Theorem 4 and 8 which use additional lemmas to bound diversity and quality terms. In addition to results for the proposed MUSS, we present new derivations tightening the known bound for DGDS with a factor of $\times 2$. Since MUSS uses both cluster and object-level selection, our bounds rely on Lemma 5 that relates

---

[1]The addition of the top $k$ items has been motivated by Lemma 7 for a tighter approximation bound. Empirically, our method achieves a similar performance with and without the top $k$ addition (Appendix C.7).

objectives at different levels. This lemma is one of our main theoretical innovation points, along with new proof approach for Lemma 1. All proofs are provided in Appendix B.

**Lemma 1.** *Apply Algorithm 1 to select* $\mathcal{S} = \text{ALG1}(\mathcal{T}|k)$. *Let* $\boldsymbol{t} \in \mathcal{T} \setminus \mathcal{S}$. *The following inequalities hold*

$$\Delta(\boldsymbol{t}, \mathcal{S}) \equiv Q(\mathcal{S} \cup \{\boldsymbol{t}\}) - Q(\mathcal{S}) \leq \frac{1}{k\lambda} F(\mathcal{S}) \tag{3}$$

$$\min_{\boldsymbol{z} \in \mathcal{S}} d(\boldsymbol{t}, \boldsymbol{z}) \leq \frac{2.5}{k(k-1)(1-\lambda)} F(\mathcal{S}). \tag{4}$$

We derive the next two lemmas enabling improved bounds for DGDS.

**Lemma 2.** *For each partition, apply Algorithm 1 to select* $\mathcal{S}_i = \text{ALG1}(\mathcal{U}_i)$. *We have*

$$D(\mathcal{O}) \leq 6F\big(\text{OPT}(\cup_{i=1}^{l}\mathcal{S}_i)\big). \tag{5}$$

**Lemma 3.** *For each partition, apply Algorithm 1 to select* $\mathcal{S}_i = \text{ALG1}(\mathcal{U}_i)$. *We have*

$$Q(\mathcal{O}) \leq 2F\big(\text{OPT}(\cup_{i=1}^{l}\mathcal{S}_i)\big). \tag{6}$$

**Theorem 4.** *With the above lemmas in place, we obtain the* $\frac{1}{16}$-*approximation solution for maximizing* $F(\mathcal{S})$ *subject to* $|\mathcal{S}| = k$

$$F\big(\text{DGDS}(\mathcal{U})\big) \geq \frac{1}{16} F(\mathcal{O}). \tag{7}$$

We now return to MUSS. Using OPT(.) to denote the selection that maximizes the objective $F$, we proceed to the following lemma.

**Lemma 5.** *Let* $k \geq m$, *we have that*

$$F\Big(\text{ALG1}\big(\mathcal{C}|m, \lambda_c, (1-\lambda_c)\big)\Big) \leq F\Big(\text{OPT}(\cup_{i=1}^{m}\mathcal{S}_i)|k\Big) + rm(m-1). \tag{8}$$

Lemma 5 connects objective functions at the cluster level and at the item level. In turn, this allows us to obtain lower bounds on the diversity term and the quality term when the multilevel Algorithm 2 is used to select $\mathcal{S} = \text{ALG2}(\mathcal{U})$.

**Lemma 6.** *If* $k \geq m$, *we have*

$$D(\mathcal{O}) \leq rk(k-1)\Big[4 + \frac{5}{1-\lambda_c}\Big] + F\Big(\text{OPT}(\cup_{i=1}^{m}\mathcal{S}_i)|k\Big)\Big[\frac{5k(k-1)}{(1-\lambda_c)m(m-1)} + \frac{1}{(1-\lambda)}\Big]. \tag{9}$$

**Lemma 7.** *Let* $\mathcal{S}^* = \arg\max_{A \subseteq \mathcal{U}, |A|=k} \sum_{\boldsymbol{u} \in A} q(\boldsymbol{u})$ *denote the set of* $k$ *highest quality items from* $\mathcal{U}$. *We have*

$$Q(\mathcal{O}) \leq \frac{1}{\lambda} F\Big(\text{OPT}\big(\cup_{i=1}^{m}\mathcal{S}_i \cup \mathcal{S}^*\big)\Big). \tag{10}$$

Finally, our main theoretical result follows.

**Theorem 8.** *In Line 7 of* ALG2, *instead of invoking* ALG1 *with* $\lambda$ *and* $1 - \lambda$, *let use parameters* $\sigma\lambda$, $1 - \lambda$. *If* $\sigma = 0.5$, $k \geq m$, $\text{ALG2}_\sigma$ *gives a constant-factor approximation to the optimal solution for maximizing* $F(\mathcal{S})$ *s.t.* $|\mathcal{S}| = k$.

$$F\big(\text{ALG2}_\sigma(\mathcal{U})\big) \geq \frac{1}{\alpha} F(\mathcal{O}) - r\frac{\beta}{\alpha}. \tag{11}$$

*Here,* $\alpha(k, m, \lambda, \lambda_c)$, $\beta(k, m, \lambda, \lambda_c)$ *are intermediate quantities defined in the proof in the interest of space.*

## 2.4 DISCUSSION

**Theoretical considerations.** In the above theorem, intermediate quantities $\alpha$ and $\beta$ are functions of algorithm parameters $k$, $m$, $\lambda$, $\lambda_c$. For fixed parameter values, $\alpha$ and $\beta$ are positive constants. In particular, if we set $k = m$ and $\lambda = \lambda_c$, we get $\alpha = 14$. This results in a better scaler compared

to Eq. (7), but our bound also has the second term as the by product of the clustering and cluster selection.

We emphasize that our theoretical analysis does not make any assumptions about the quality of clustering. Instead, we note that the bound improves as $r$ gets smaller. Growing the number of clusters $l$ will make this radius smaller, but will increase time required for selecting clusters (Table 1). The ideal case is when the data naturally forms a small number of clusters, such that $l$ and $r$ are both low.

Next, the bound can be explicitly maximized as a function of $m$ and $\lambda_c$. However, in practice, we simply evaluate results for different values of $\lambda_c$, while $m$ is selected to balance objective value with computational time.

Lastly, note that parameters $k$ and $\lambda$ are included in the objective function (Eq. 2). However, these parameters are application-driven, and should not be used to "optimize" the approximation bound. E.g., for a given application, the best $\lambda$ value is the one that results in strongest correlation between an application-specific performance metric and the objective $F$.

**Practical considerations.** One of the benefits of the proposed approach is that clustering can be performed in advance at a preprocessing stage. Each time a selection is required, a pre-existing clustering structure is leveraged. For large datasets, one can use scalable clustering methods, such as MiniBatchKmeans (Sculley, 2010) or FAISS (Douze et al., 2024). If new data arrives, an online clustering update can be used. In a simple case, one can store pre-computed cluster centroids and assign each newly arriving point to the nearest center.

In practice, we use the same parameter $\lambda$ when selecting items either within clusters (Line 5 of Algorithm 2) or from the union of selections (Line 7 of Algorithm 2). However, our method is flexible, and one can consider different $\lambda$ values for these selection stages. Next, during the greedy selection, we normalize the sum of distances by the current selection size $|\mathcal{S}|$ for robustness.

Theorem 8 assumes $\sigma = 0.5$, i.e., a scaler in Line 7 of the Algorithm 2. Importantly, the approximation bound still holds when running the algorithm with different values of $\sigma$ and selecting result that maximizes $F$. Indeed, our preliminary results indicated that using $\sigma = 1$ (i.e., no scaling) leads to a stronger performance. Therefore MUSS is defined without the scaler. Also note that the original DGDS baseline does use scaling by $0.5$. In our evaluation, removing this scaling resulted in better DGDS results which we report here.

**Benefits of cluster selection.** Since item selection within clusters can be performed in parallel, the main performance bottleneck is item selection from the union of subsets derived from different clusters. To reduce the size of this union, we introduce a novel idea of relevant and diverse selection of clusters. This step can dramatically reduce the number of items at the final selection with minimum impact on the selection quality. To the best of our knowledge, previous approaches did not consider the idea of "pruning" the set of clusters.

Preliminary elimination of a large number of clusters (Line 3 of Algorithm 2) will not only allow for more efficient selection from the union of points (running time and memory for Line 7 of Algorithm 2), but can lead to improved accuracy. This is because the greedy algorithm will be able to focus on relevant items after redundancy across clusters has been reduced. This is particularly useful for large scale dataset size, as shown in our experiments. Moreover, novel theoretical analysis, such as Lemma 5, allows us to relate cluster-level and item-level selection stages and derive an approximation bound for the proposed MUSS.

## 3 EXPERIMENTS

The goals of our experiments have been to (i) test whether the proposed MUSS can be useful in practical applications; (ii) understand the impact of different components of our method, and (iii) understand scalability and parameter sensitivity of the proposed approach. Item recommendation and retrieval-augmented generation are among the most prominent applications of our subset selection problem. In the next two sections, we consider these applications, and compare MUSS with a number of baselines.

**Baselines.** We consider the following methods for the task of high quality and diverse subset selection: random selection, K-DPP (Kulesza & Taskar, 2011), clustering-based selection, MMR as per

Table 2: Precision on the candidate retrieval task for $k = 500$ items. ✗ indicates that the method did not complete after 12 hours. Results are reported for $\lambda$ that maximizes precision achieved by MMR (i.e., favoring the baseline). For any value of $\lambda_c$, our method achieves higher performance than baselines and faster running time.

| Home ($|\mathcal{U}| = 4737, \lambda = 0.9$) | | | | Amazon100k ($|\mathcal{U}| = 108,258, \lambda = 0.9$) | | | |
|---|---|---|---|---|---|---|---|
| **Method** | $\lambda_c$ | **Precision** ↑ | **Time** ↓ | **Method** | $\lambda_c$ | **Precision** ↑ | **Time** ↓ |
| random | | $50.3 \pm 2.4$ | 0.0 | random | | $11.2 \pm 1.5$ | 0.0 |
| K-DPP | | $56.3 \pm 2.7$ | 7.9 | K-DPP | | ✗ | ✗ |
| clustering | | $60.6 \pm 1.8$ | 0.7 | clustering | | $28.2 \pm 1.1$ | 10 |
| MMR | | 72.0 | 13.5 | MMR | | 39.4 | 311 |
| DGDS | | $73.5 \pm 0.2$ | 13.7 | DGDS | | $39.4 \pm 0.1$ | 271 |
| MUSS (rand.A) | | $73.9 \pm 0.3$ | 6.7 | MUSS (rand.A) | | $42.8 \pm 0.3$ | 49 |
| MUSS (rand.B) | | $74.1 \pm 0.2$ | 6.6 | MUSS (rand.B) | | $41.6 \pm 0.2$ | 53 |
| MUSS | 0.1 | $74.5 \pm 0.2$ | 7.1 | MUSS | 0.1 | $44.8 \pm 0.5$ | 55 |
| MUSS | 0.3 | $74.2 \pm 0.3$ | 7.8 | MUSS | 0.2 | $42.8 \pm 0.8$ | 54 |
| MUSS | 0.5 | $74.0 \pm 0.3$ | 7.8 | MUSS | 0.5 | $43.5 \pm 0.5$ | 54 |
| MUSS | 0.7 | $74.1 \pm 0.3$ | 8.8 | MUSS | 0.7 | $44.4 \pm 0.4$ | 53 |
| MUSS | 0.9 | $\mathbf{74.8 \pm 0.2}$ | 8.1 | MUSS | 0.9 | $\mathbf{45.2 \pm 0.6}$ | 53 |

Algorithm 1, and the distributed selection method called DGDS (Ghadiri & Schmidt, 2019). We do not consider RL baselines here because we focus on selection methods that are potentially scalable, and also can be easily incorporated within existing ML systems. RL-based selection approaches require setting up a feedback loop and defining rewards which might not be trivial in a given ML application.

Key differences between DGDS and MUSS are that (i) we propose clustering rather than random partitioning, (ii) we select a subset of clusters, rather than using all of them (iii) in the final selection, MUSS takes into account the top $k$ highest quality items while DGDS does not. To understand the impact of these differences, we introduce two additional variations of our method. First, in "MUSS (rand.A)", we perform clustering, but pick $m$ clusters at random rather than using greedy selection. Second, in "MUSS (rand.B)", we perform random partitioning instead of clustering, but otherwise follow our Algorithm 2.

We report mean± st.err. from 5 independent runs. In each run, randomness is due to partitioning, clustering or sampling (K-DPP). There are no repeated runs for MMR, since this method doesn't use partitioning nor randomness. Additional experimental details are given in Appendix C.1.

## 3.1 CANDIDATE RETRIEVAL FOR PRODUCT RECOMMENDATION

**Context.** Modern recommender systems typically consist of two stages. First, candidate retrieval aims at efficiently identifying a subset of relevant items from a large catalog of items (El-Kishky et al., 2023; Rajput et al., 2023). This step narrows down the input space for the second, more expensive, ranking stage. Since the ranking will not even consider items missed by candidate retrieval, it is crucial for the candidate retrieval stage to maximize recall — ensuring that most relevant items are included in the retrieved subset — while maintaining computational efficiency. The proposed MUSS has been deployed in production at a large-scale ecommerce platform serving million customers daily, referring to Appendix C.2 for further information.

**Setup.** We use four datasets with sizes ranging from 4K to 2M (Table 2 and Appendix Table 6). These internally collected datasets represent either individual product categories, or larger collections of items across categories. Each data point corresponds to a product available at an online shopping service. For each product, an external ML model predicts the likelihood of an item being clicked on. The model takes into account product attributes, embedding, and historical performance. Likelihood predictions are treated as product quality scores, while actual clicks data is used as binary labels. We select $k = 500$ items from a given dataset. For a fixed $k$ recall is proportional to precision@k, and we evaluate selection performance using Precision@500.

**Results** are shown in Table 2, and Appendix Table 6. First, higher values of the objective from Eq. (2) generally indicate higher precision, which further justifies our problem formulation. Next, it is clear that random selection or naive clustering-based strategy are not effective for this task as

Table 3: Accuracy of question answering over different knowledge bases given a fixed LLM, but varying methods for RAG selection. $\lambda$ values were optimized on MMR accuracy, thus favoring this baseline. For MUSS (rand.B) variation we use $\lambda_c$ that maximized performance of this method. MUSS outperforms all baselines. We are interested in accuracy rather than timing, since the response time is dominated by the LLM call.

| DevOps ($|\mathcal{U}| = 4722$, $\lambda = 0.5$) | | | StackExchange ($|\mathcal{U}| = 1025$, $\lambda = 0.5$) | | |
|---|---|---|---|---|---|
| **Method** | $\lambda_c$ | **Accuracy** ↑ | **Method** | $\lambda_c$ | **Accuracy** ↑ |
| random | | $50.0 \pm 1.1$ | random | | $41.6 \pm 2.0$ |
| K-DPP | | $47.6 \pm 1.8$ | K-DPP | | $40.4 \pm 1.5$ |
| clustering | | $51.2 \pm 0.5$ | clustering | | $54.8 \pm 4.4$ |
| MMR | | 58 | MMR | | 64 |
| DGDS | | $58.0 \pm 0.0$ | DGDS | | $62.8 \pm 0.5$ |
| MUSS (rand.A) | | $52.0 \pm 2.2$ | MUSS (rand.A) | | $55.2 \pm 4.8$ |
| MUSS (rand.B) | | $53.2 \pm 2.1$ | MUSS (rand.B) | | $55.6 \pm 4.7$ |
| MUSS | 0.1 | $58.8 \pm 0.5$ | MUSS | 0.1 | $65.2 \pm 0.8$ |
| MUSS | 0.3 | $58.8 \pm 1.0$ | MUSS | 0.3 | $65.2 \pm 0.8$ |
| MUSS | 0.5 | $58.8 \pm 0.5$ | MUSS | 0.5 | $\mathbf{65.6 \pm 1.0}$ |
| MUSS | 0.7 | $\mathbf{59.6 \pm 0.7}$ | MUSS | 0.7 | $64.8 \pm 0.8$ |
| MUSS | 0.9 | $58.0 \pm 0.6$ | MUSS | 0.9 | $64.8 \pm 0.5$ |

all other methods significantly outperform these baselines. Here, we use $\lambda = 0.9$ which maximizes precision resulted from using MMR. Even with this $\lambda$ choice, MUSS achieves consistently higher precision ($+4\%$) across various $\lambda_c$ values. This improvement is due to the property of MUSS to perform selection within each subgroup, allowing the selection process to better capture local structure and diversity specific to each subgroup than handling all items globally. Importantly, MUSS achieves this results $80\times$ faster than MMR (Amazon2M) and $35\%$ faster than DGDS. Improved scalability can be observed on datasets of different sizes (Figure 1 and Appendix Figure 7).

## 3.2 Q&A USING RETRIEVAL-AUGMENTED GENERATION

**Context.** Recently, Large Language Models (LLM) have gained significant popularity as core methods for a range of applications, from question answering bots to code generation. Retrieval-augmented Generation (RAG) refers to a technique where information relevant to the task is retrieved from a knowledge base and added to the LLM's prompt. Given the importance of RAG, we have also evaluated MUSS for RAG entries selection.

**Setup.** We consider the task of answering questions over a custom knowledge corpus, and we use two datasets of varying degrees of difficulty (Table 3). StackExchange and DevOps datasets represent more specialized knowledge.[2] These datasets were derived, respectively, from an online technical question answering service, and from AWS Dev Ops troubleshooting pages (Guinet et al., 2024).

Each dataset consists of a knowledge corpus and a number of multiple choice questions. For a given question, we compute relevance to entities in the corpus, and then use different methods for selecting $k = 3$ relevant and diverse entities to be added to LLM's prompt. For a fixed LLM we vary selection methods, and report proportion of correct answers over 50 questions.

In this section, we are interested in accuracy of the answers rather than timing. We assume that given a question, one can effectively narrow down relevant scope of knowledge and the response time might be dominated by the LLM call.

**Results** are presented in Table 3. In all cases, accuracy can be improved compared to random selection. Parameter $\lambda$ (item-level selection trade-off) is optimized for MMR performance, thus favoring this baseline. Maximum accuracy is achieved with an intermediate value of the parameter, i.e., both relevance and diversity are important. Random selection and K-DPP baselines emphasize diversity over relevance and achieve the weakest performance.

We can see that our method is capable of outperforming all baselines, particularly at any $\lambda_c$ value. Note that the two datasets involve complex technical questions, and RAG approach itself might stop

---

[2]https://github.com/amazon-science/auto-rag-eval

Table 4: Precision (for Home and Amazon100k) or Accuracy (other datasets) achieved by MUSS with different number of clusters $l$, number of selected clusters $m$, and fixed $\lambda = \lambda_c = 0.7$.

| $l$ | $m$ | **Home** | **Amazon100k** | $l$ | $m$ | **DevOps** | **StackExchange** |
|-----|-----|----------|----------------|-----|-----|------------|-------------------|
| 100 | 50  | 74.6     | 41.6           | 50  | 10  | 46         | 62                |
| 200 | 50  | 74.8     | 41.2           | 50  | 20  | 46         | 62                |
| 200 | 100 | 74.8     | 44.0           | 100 | 10  | 44         | 62                |
| 500 | 50  | 73.3     | 42.8           | 100 | 20  | 46         | 62                |
| 500 | 100 | 74.0     | 44.2           | 200 | 10  | 46         | 62                |
| 500 | 200 | 74.2     | 44.2           | 200 | 20  | 44         | 62                |

being effective past certain performance level. Nonetheless, our findings suggest that as long as RAG continues to contribute to performance gains, our method can further enhance accuracy.

### 3.3 ABLATIONS, PARAMETER SENSITIVITY, AND SCALABILITY

**Ablation Study.** Note that variations "MUSS (rand.A)", and "MUSS (rand.B)" constitute ablations of our method. In the former, we select clusters at random instead of using cluster-level greedy selection. We observe that using greedy selection consistently improves performance. Next, in "MUSS (rand.B)", we use random partitioning instead of clustering. Again, we consistently observe improved performance when clustering is applied, and the gains can be significant. We conclude that leveraging natural structure in data is important for this problem. This is consistent with observed patterns discussed in Appendix C.3.

**Sensitivity w.r.t. $\lambda$ and $\lambda_c$.** Table 2, Table 3, and Appendix Table 6 show performance at different levels of $\lambda_c$ (cluster-level trade-off). Overall, for any dataset, there is little variation in performance. We also study how the diversity term $D(\mathcal{S})$, the quality term $Q(\mathcal{S})$, and the objective function $F(\mathcal{S})$ varies with $\lambda$ and $\lambda_c$ in Appendix C.4. Consistent with the previous observation, we find that for any fixed $\lambda$, the variation due to $\lambda_c$ is relatively small. Next, as expected, small values of $\lambda$ (item-level trade-off) favor $D(\mathcal{S})$ while larger $\lambda$ promote $Q(\mathcal{S})$. The optimal choice of this parameter is application-specific. A practical way of setting the value could be cross-validation at some fixed $\lambda_c$.

**Sensitivity w.r.t. number of clusters $l$ and number of selected clusters $m$.** We consider broad ranges for these parameter values. For example, we scale $l$ by 4 to 5 times, and $m$ by 2 to 4 times (while keeping both $\lambda$s fixed). Despite broad parameter ranges, in most cases, performance differences between different settings are within 3 percent points (Table 4). Larger deviations are typically observed as settings become more extreme (e.g., number of clusters is becoming too little for a dataset with 100k items).

**Scalability.** Figure 1 demonstrates scalability of the proposed MUSS. Specifically, given the dataset of size $|\mathcal{U}| = 2M$, our method is up to 80 times faster than MMR achieving the same objective function of 0.97. Here, all methods use the same $\lambda = 0.5$ and we fix the hyperparameters to some constant values ($m = 100, l = 500, \lambda_c = 0.5$). Further analysis into scalability shows that compared to DGDS, our approach leads to time savings both during selection within partitions and during the final selection from the union of items (Appendix C.5 and Appendix Figure 7).

## 4 CONCLUSION

We propose a novel method for distributed relevant and diverse subset selection. We complement our method with theoretical analysis that relates cluster- and item-level selection and enables us to derive an approximation bound. Our evaluation shows that the proposed MUSS can considerably outperform baselines both in terms of scalability and performance on the target applications. The problem of relevant and diverse subset selection has a wide range of applications, e.g., recommender systems and retrieval-augmented generation (RAG). This problem is NP-hard, and popular approaches such as Maximum Marginal Relevance (MMR) are based on greedy selection. Later methods, such as DGDS considered a distributed setting using random data partitioning. In contrast, in our work, we leverage clustering structure in the data for better performance. Finally, the proposed MUSS has been deployed in production on a large-scale e-commerce retail platform.

## REPRODUCIBILITY STATEMENT

To ensure the reproducibility of our work, we provide comprehensive implementation details and experimental protocols throughout the paper and appendices. Due to the simplicity nature of the proposed method, all algorithms are fully specified: Algorithm 2 details MUSS implementation and Algorithm 1 describes the MMR selection – as one of the key step inside MUSS.

All hyperparameters and computational configuration are specified in Appendix C.1. The public datasets used for RAG experiment are described in Section 3.2 and the datasets used for candidate retrieval tasks are described in Section C.2.

The mathematical foundations, including all proofs for Theorems and Lemmas are provided in Appendix B. The computational runtimes are shown in Fig. 7. We attach the source code in the supplementary material. To support transparency and broader use, we will release this code and evaluation scripts to Github upon publication, enabling full reproducibility of the reported results.

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

# A RELATED WORK

## A.1 RELEVANT AND DIVERSE SELECTION

Given the importance of the problem, there has been a number of approaches proposed in the literature. **Determinantal Point Process (DPP)** is a probabilistic model that selects diverse subsets by maximizing the determinant of a kernel matrix representing item similarities (Kulesza et al., 2012). DPPs are effective in summarization, recommendation, and clustering tasks (Wilhelm et al., 2018; Elfeki et al., 2019; Yuan & Kitani, 2020; Nguyen et al., 2021). As discussed in reference (Li et al., 2016; Derezinski et al., 2019), the computational complexity of k-DPP can be $\mathcal{O}(k^2 n + k^3)$. Next, **clustering-based methods** ensure that different "regions" of the dataset are covered by the selection. Such methods cluster items (e.g., documents or features) and then select representatives from each cluster (Baeza-Yates, 2005; Wang et al., 2021; Panteli & Boutsinas, 2023; Ge et al., 2024). This approach is commonly used in text and image summarization. We use clustering in our method, but we depart from previous work in many other aspects (e.g., how we select within clusters, pruning clusters, theoretical analysis).

**Reinforcement learning (RL)** frameworks can be used to optimize diversity and relevance in sequential tasks such as recommendation and active learning. However, achieving an optimal balance between exploring diverse solutions and exploiting high-quality ones can be challenging, often leading to suboptimal convergence or increased training time (Levine et al., 2020; Fontaine & Nikolaidis, 2021). **Model-based methods** use application-specific probabilistic models or properties of relevance, quality, and diversity (Gao & Zhang, 2024; Pickett et al., 2024; Hirata et al., 2022; Acharya et al., 2024).

**Maximum Marginal Relevance (MMR)** is one of the most popular approaches for balancing relevance and diversity (Carbonell & Goldstein, 1998). Effectiveness of MMR has been demonstrated in numerous studies (Erkan & Radev, 2004; Wan & Yang, 2008; Xia et al., 2015). The algorithm was introduced in the context of retrieving similar but non-redundant documents for a given query $q$. Let $\mathcal{U}$ denote document corpus and $\mathcal{S}$ denote items selected so far. In each iteration, MMR evaluates all remaining candidates and selects item $s \in \mathcal{U} \setminus \mathcal{S}$ that maximizes criterion:

$$\text{MMR}(s) = \lambda \cdot \text{Sim}(s, q) - (1 - \lambda) \cdot \max_{t \in \mathcal{S}} \text{Sim}(s, t),$$

where $\text{Sim}(.,.)$ measures similarity between two items, and $\lambda$ controls the trade-off between relevance and diversity.

## A.2 DISTRIBUTED GREEDY SELECTION

The problem of subset selection can be viewed as maximization of a set-valued objective that assigns high values to subsets with desired properties (e.g., relevance of elements). Submodular functions is a special class of such objectives that has attracted significant attention. In particular, for a non-negative, monotone submodular function $f : 2^{\mathcal{U}} \to \mathbb{R}$ and a cardinality constraint $k$, the solution $\mathcal{S}_g$ obtained by the greedy algorithm satisfies: $f(\mathcal{S}_g) \geq \left(1 - \frac{1}{e}\right) f(\mathcal{S}^*)$ where $\mathcal{S}^*$ is the optimal solution of size at most $k$ (Nemhauser et al., 1978).

**Distributed submodular maximization** is an approach to solve submodular optimization problems in a distributed manner, e.g., when the dataset is too large to handle on a single machine (Mirzasoleiman et al., 2016; Barbosa et al., 2015). The authors provide theoretical analysis showing that under certain conditions one can achieve performance close to the non-distributed approach.

Since the addition of the diversity requirement results in a non-submodular objective for relevant and diverse selection, researchers had to relax the requirement for submodularity.

**Beyond submodular maximization** Ghadiri and Schmidt consider distributed maximization of so-called "submodular plus diversity" functions (Ghadiri & Schmidt, 2019). The authors introduce a framework, called DGDS, for multi-label feature selection that balances relevance and diversity in the context of large-scale datasets. Their work addresses computational challenges posed by traditional submodular maximization techniques when applied to high-dimensional data. The authors propose a distributed greedy algorithm that leverages the additive structure of submodular plus diversity functions. This framework enables the decomposition of the optimization problem across multiple computational nodes, significantly reducing running time while preserving effectiveness.

Table 5: Notation used throughout the paper.

| Variable | Definition |
|---|---|
| $\boldsymbol{u}_i = (\boldsymbol{x}_i, q_i)$ | an item as a pair (embedding $\boldsymbol{x}_i \in \mathbb{R}^d$, quality score $q_i \in \mathbb{R}^+$) |
| $\mathcal{U} = \{\boldsymbol{u}_i\}_{i=1}^n$ | universe of items, dataset of size $n$ from which we select items |
| $\mathcal{U}_1, \ldots, \mathcal{U}_m, \ldots, \mathcal{U}_l$ | partitioned data, i.e., $\cup_{i=1}^l \mathcal{U}_i = \mathcal{U}$ |
| $r \geq 0$ | maximum radii from an item to its cluster centre |
| $p \in \mathbb{N}$ | the number of CPUs or computational threads for parallel jobs |
| $\mathcal{S} \subseteq \mathcal{U}, k$ | a set of selected items; number of items to select, $|\mathcal{S}| = k$ |
| $\mathcal{C}, l, m$ | a set of clusters (partitions); # clusters; # clusters to be selected, $l \geq m$. |
| $0 \leq \lambda, \lambda_c \leq 1$ | trade-off parameters between quality and diversity at different levels |
| $\bar{\mathcal{S}} = \text{ALG1}(\mathcal{C}|m)$ | $m$ clusters selected from $\mathcal{C}$ using Algorithm 1 |
| $\mathcal{S}_i = \text{ALG1}(\mathcal{U}_i|k)$ | $k$ items selected from $\mathcal{U}_i$ using Algorithm 1 |
| $Q(\mathcal{S}), D(\mathcal{S})$ | quality and diversity of subset $\mathcal{S}$ |
| $\Delta(\boldsymbol{t}, \mathcal{S}) = Q(\mathcal{S} \cup \{\boldsymbol{t}\}) - Q(\mathcal{S})$ | gain in quality score of subset $\mathcal{S}$ resulted from adding $\boldsymbol{t}$ to this subset |

However, items selected from different partitions are ultimately combined to perform the final selection step. Selecting objects in each partition, along with the final selection step becomes a performance bottleneck. Therefore, we further improve scalability of distributed selection by exploiting natural clustering structure in the data (Table 1).[3] Moreover, we complement our method with a novel theoretical analysis of clustering-based selection.

## B    PROOFS OF LEMMAS AND THEOREMS

In this appendix, we present proofs of lemmas and the theorem that represents our main result. Throughout the proofs, we make technical assumptions that $k > 1$, $\lambda \neq 0$, and $\lambda \neq 1$ to avoid zero denominators. Key notation used throughout the paper is summarized in Appendix Table 5.

### B.1    PROOF OF LEMMA 1

*Proof.* Let ALG1 denote Algorithm 1. For any $\boldsymbol{z} \in \mathcal{U}$ and $\mathcal{S} \subseteq \mathcal{U}$, let $\Delta(\boldsymbol{z}, \mathcal{S}) := Q(\mathcal{S} \cup \{\boldsymbol{z}\}) - Q(\mathcal{S})$. Next, let $\boldsymbol{z}_1, \ldots, \boldsymbol{z}_k$ denote items that the algorithm ALG1 selected in the order of selection. Define $\mathcal{S}_i := \{\boldsymbol{z}_1, \ldots, \boldsymbol{z}_i\}$ and $\mathcal{S}_0 := \emptyset$. Finally, let $\boldsymbol{t} \in \mathcal{T} \backslash \text{ALG1}(\mathcal{T}|k, \lambda)$.

Due to the greedy selection mechanism, we have the following

$$\lambda\Delta(\boldsymbol{z}_1, \mathcal{S}_0) \geq \lambda\Delta(\boldsymbol{t}, \mathcal{S}_0) \tag{12}$$

$$\lambda\Delta(\boldsymbol{z}_2, \mathcal{S}_1) + (1-\lambda)d(\boldsymbol{z}_2, \boldsymbol{z}_1) \geq \lambda\Delta(\boldsymbol{t}, \mathcal{S}_1) + (1-\lambda)d(\boldsymbol{t}, \boldsymbol{z}_1) \tag{13}$$

$$\cdots$$

$$\lambda\Delta(\boldsymbol{z}_k, \mathcal{S}_{k-1}) + (1-\lambda)\sum_{i=1}^{k-1} d(\boldsymbol{z}_k, \boldsymbol{z}_i) \geq \lambda\Delta(\boldsymbol{t}, \mathcal{S}_{k-1}) + (1-\lambda)\sum_{i=1}^{k-1} d(\boldsymbol{t}, \boldsymbol{z}_i). \tag{14}$$

Adding these inequalities together gives us

$$\lambda Q(\mathcal{S}_k) + \frac{(1-\lambda)}{2}D(\mathcal{S}_k) \geq (1-\lambda)\sum_{i=1}^{k-1}(k-i)d(\boldsymbol{t}, \boldsymbol{z}_i) + \lambda\sum_{i=0}^{k-1}\Delta(\boldsymbol{t}, \mathcal{S}_i). \tag{15}$$

Since $(1-\lambda)D(\mathcal{S}_k) \geq \frac{(1-\lambda)}{2}D(\mathcal{S}_k)$, we have

$$\lambda Q(\mathcal{S}_k) + (1-\lambda)D(\mathcal{S}_k) \geq (1-\lambda)\sum_{i=1}^{k-1}(k-i)d(\boldsymbol{t}, \boldsymbol{z}_i) + \lambda\sum_{i=0}^{k-1}\Delta(\boldsymbol{t}, \mathcal{S}_i) \tag{16}$$

$$F(\mathcal{S}_k) \geq (1-\lambda)\sum_{i=1}^{k-1}(k-i)d(\boldsymbol{t}, \boldsymbol{z}_i) + \lambda k\Delta(\boldsymbol{t}, S_k) \tag{17}$$

---

[3]For particular relevance and diversity definitions, complexity of greedy selection used in MMR, DGDS, and MUSS can be reduced to $\mathcal{O}(kn)$, but the main benefit of MUSS, which is reducing dependency on $n$, still applies.

where the second inequality is due to submodularity of $Q$. This immediately gives $\Delta(\boldsymbol{t}, \mathcal{S}_k) \leq \frac{1}{k\lambda} F(\mathcal{S}_k)$ which concludes the first part of the Lemma.

Next, introduce intermediate quantities $T_A = \sum_{i=1}^{k-1} (k-i) d(\boldsymbol{t}, \boldsymbol{z}_i)$ and $T_B = \sum_{i=2}^{k} (i-1) d(\boldsymbol{t}, \boldsymbol{z}_i)$.

Since $d(.,.)$ is a metric, we have the triangle inequalities

$$d(\boldsymbol{t}, \boldsymbol{z}_k) \leq d(\boldsymbol{z}_k, \boldsymbol{z}_1) + d(\boldsymbol{t}, \boldsymbol{z}_1)$$
$$d(\boldsymbol{t}, \boldsymbol{z}_k) \leq d(\boldsymbol{z}_k, \boldsymbol{z}_2) + d(\boldsymbol{t}, \boldsymbol{z}_2)$$
$$d(\boldsymbol{t}, \boldsymbol{z}_k) \leq d(\boldsymbol{z}_k, \boldsymbol{z}_3) + d(\boldsymbol{t}, \boldsymbol{z}_3)$$
$$\ldots$$
$$d(\boldsymbol{t}, \boldsymbol{z}_k) \leq d(\boldsymbol{z}_k, \boldsymbol{z}_{k-1}) + d(\boldsymbol{t}, \boldsymbol{z}_{k-1})$$
$$\ldots$$
$$d(\boldsymbol{t}, \boldsymbol{z}_{k-1}) \leq d(\boldsymbol{z}_{k-1}, \boldsymbol{z}_1) + d(\boldsymbol{t}, \boldsymbol{z}_1)$$
$$d(\boldsymbol{t}, \boldsymbol{z}_{k-1}) \leq d(\boldsymbol{z}_{k-1}, \boldsymbol{z}_2) + d(\boldsymbol{t}, \boldsymbol{z}_2)$$
$$\ldots$$
$$d(\boldsymbol{t}, \boldsymbol{z}_2) \leq d(\boldsymbol{z}_2, \boldsymbol{z}_1) + d(\boldsymbol{t}, \boldsymbol{z}_1) \tag{18}$$

Adding these inequalities together gives $T_B \leq \frac{1}{2} D(\mathcal{S}_k) + T_A$.

We plug this result into Eq. (17) to have

$$F(\mathcal{S}_k) \geq (1 - \lambda) T_A \tag{19}$$
$$F(\mathcal{S}_k) + (1 - \lambda) T_B \geq (1 - \lambda) T_A + (1 - \lambda) T_B \tag{20}$$
$$F(\mathcal{S}_k) + \frac{1 - \lambda}{2} D(\mathcal{S}_k) + (1 - \lambda) T_A \geq (1 - \lambda) T_A + (1 - \lambda) T_B \tag{21}$$
$$F(\mathcal{S}_k) + \frac{1 - \lambda}{2} D(\mathcal{S}_k) + F(\mathcal{S}_k) \geq (1 - \lambda) T_A + (1 - \lambda) T_B \tag{22}$$
$$2.5 F(\mathcal{S}_k) \geq (1 - \lambda) T_A + (1 - \lambda) T_B \tag{23}$$
$$2.5 F(\mathcal{S}_k) \geq (1 - \lambda)(k - 1) \sum_{i=1}^{k} d(\boldsymbol{t}, \boldsymbol{z}_i) \tag{24}$$
$$\frac{2.5}{k - 1} F(\mathcal{S}_k) \geq (1 - \lambda) \sum_{i=1}^{k} d(\boldsymbol{t}, \boldsymbol{z}_i) \tag{25}$$

where we apply Eq. (19) to obtain Eq. (22). We utilize $T_A + T_B = (k - 1) \sum_{i=1}^{k} d(\boldsymbol{t}, \boldsymbol{z}_i)$ in Eq. (24).

Finally, we have that

$$\frac{2.5}{k(k - 1)} F(\mathcal{S}_k) \geq (1 - \lambda) \frac{1}{k} \sum_{i=1}^{k} d(\boldsymbol{t}, \boldsymbol{z}_i) \geq (1 - \lambda) \min_{i=1,\ldots,k} d(\boldsymbol{t}, \boldsymbol{z}_i). \tag{26}$$

This is because the minimum of positive values is not greater than their average. This concludes the proof.

The same way as ALG1 can be used for selection of both clusters and individual items, this Lemma applies at both cluster and individual item levels. ∎

### B.2 PROOF OF LEMMA 2

*Proof.* Let $h(\boldsymbol{u})$ denote a mapping where each data point $\boldsymbol{u} \in \mathcal{O} \cap \mathcal{U}_i$ is mapped to the nearest selected point from the same partition, thus $h(\boldsymbol{u}) \in \mathcal{S}_i$. Note that for points already in $\mathcal{O} \cap \mathcal{S}_i$ this is the identity mapping.

Since $d(.,.)$ is a metric, we have

$$D(\mathcal{O}) = \sum_{\boldsymbol{u},\boldsymbol{v}\in\mathcal{O}} d(\boldsymbol{u},\boldsymbol{v}) \tag{27}$$

$$\leq \sum_{\boldsymbol{u}\in\mathcal{O}} \sum_{\boldsymbol{v}\in\mathcal{O},\boldsymbol{v}\neq\boldsymbol{u}} \Big( d\big(\boldsymbol{u},h(\boldsymbol{u})\big) + d\big(\boldsymbol{v},h(\boldsymbol{v})\big) + d\big(h(\boldsymbol{u}),h(\boldsymbol{v})\big) \Big) \tag{28}$$

$$= (k-1) \sum_{\boldsymbol{u}\in\mathcal{O}} d\big(\boldsymbol{u},h(\boldsymbol{u})\big) + (k-1) \sum_{\boldsymbol{v}\in\mathcal{O}} d\big(\boldsymbol{v},h(\boldsymbol{v})\big) + \sum_{\boldsymbol{u},\boldsymbol{v}\in\mathcal{O}} d\big(h(\boldsymbol{u}),h(\boldsymbol{v})\big). \tag{29}$$

Consider the first term. For any point $\boldsymbol{u} \in \mathcal{O} \cap \mathcal{U}_i$, if $\boldsymbol{u} \in \mathcal{O} \cap \mathcal{S}_i$ then $d(\boldsymbol{u},h(\boldsymbol{u})) = 0$. Else, according to Lemma 1 we have that $d(\boldsymbol{u},h(\boldsymbol{u})) \leq \frac{2.5}{k(k-1)}F(\mathcal{S}_i)$. Thus, the first term is bounded by $2.5F(\cup_{i=1}^{l}S_i)$. The same argument applies to the second term.

Finally, consider the last term. By definition of mapping $h(.)$, we have that $h(\boldsymbol{u}) \in \cup_{i=1}^{l}S_i$ for any $\boldsymbol{u}$. Thus we have $\sum_{\boldsymbol{u},\boldsymbol{v}\in\mathcal{O}} d(h(\boldsymbol{u}),h(\boldsymbol{v})) \leq D\big(\cup_{i=1}^{l}S_i\big) \leq F\big(\cup_{i=1}^{l}S_i\big)$.

We conclude that

$$D(\mathcal{O}) \leq 6F\big(\cup_{i=1}^{l}S_i\big) \leq 6F\big(\text{OPT}(\cup_{i=1}^{l}\mathcal{S}_i)\big). \tag{30}$$

$\blacksquare$

### B.3 Proof of Lemma 3

*Proof.* Denote $\Delta(q,\mathcal{S}) = Q(\mathcal{S} \cup \{q\}) - Q(\mathcal{S})$. Let $o_1,\ldots,o_k$ be an ordering of elements of the optimal set $\mathcal{O}$. For $\boldsymbol{z} = o_i \in \mathcal{O}$ define $O_{\boldsymbol{z}} = \{o_1,\ldots,o_i - 1\}$ and $\mathcal{O}_{o_1} = \emptyset$. Finally, recall that $\mathcal{U}_i$ denotes a data partition, and $\mathcal{S}_i = \text{ALG1}\big(\mathcal{U}_i\big)$.

We bound the quality term by decomposing the optimal set $\mathcal{O}$ into points being selected and points not being selected.

$$Q(\mathcal{O}) = Q\Big(\mathcal{O} \cap (\cup_{i=1}^{l}\mathcal{S}_i)\Big) + \sum_{\boldsymbol{z}\in\mathcal{O}\setminus(\cup_{i=1}^{l}\mathcal{S}_i)} \Delta\Big(\boldsymbol{z},\mathcal{O}_{\boldsymbol{z}} \cup \big(\mathcal{O} \cap (\cup_{i=1}^{l}\mathcal{S}_i)\big)\Big) \tag{31}$$

$$\leq F\Big(\text{OPT}(\cup_{i=1}^{l}\mathcal{S}_i)\Big) + \sum_{\boldsymbol{z}\in\mathcal{O}\setminus(\cup_{i=1}^{l}\mathcal{S}_i)} \Delta(\boldsymbol{z},\mathcal{O}_{\boldsymbol{z}}) \tag{32}$$

$$= F\Big(\text{OPT}(\cup_{i=1}^{l}\mathcal{S}_i)\Big) + \sum_{i=1}^{l} \sum_{\boldsymbol{z}\in\mathcal{O}\cap\mathcal{U}_i\setminus\mathcal{S}_i} \Delta(\boldsymbol{z},\mathcal{O}_{\boldsymbol{z}} \cup \mathcal{S}_i) + \Delta(\boldsymbol{z},\mathcal{O}_{\boldsymbol{z}}) - \Delta(\boldsymbol{z},\mathcal{O}_{\boldsymbol{z}} \cup \mathcal{S}_i) \tag{33}$$

$$\leq F\Big(\text{OPT}(\cup_{i=1}^{l}\mathcal{S}_i)\Big) + \sum_{i=1}^{l} \sum_{\boldsymbol{z}\in\mathcal{O}\cap\mathcal{U}_i\setminus\mathcal{S}_i} \frac{1}{k}F(\mathcal{S}_i) \tag{34}$$

$$\leq 2F\Big(\text{OPT}(\cup_{i=1}^{m}\mathcal{S}_i)\Big). \tag{35}$$

In Eq. (34), we use the fact that $\Delta(\boldsymbol{z},\mathcal{O}_{\boldsymbol{z}}) - \Delta(\boldsymbol{z},\mathcal{O}_{\boldsymbol{z}}\cup\mathcal{S}_i) = Q(\boldsymbol{z}) - Q(\boldsymbol{z}) = 0$ and $\Delta(\boldsymbol{z},\mathcal{O}_{\boldsymbol{z}}\cup\mathcal{S}_i) \leq \Delta(\boldsymbol{z},\mathcal{S}_i)$ and also apply Lemma 1. $\blacksquare$

### B.4 Proof of Theorem 4

*Proof.* Recall that $F(\mathcal{O}) = D(\mathcal{O}) + Q(\mathcal{O})$. Using new results from Lemma 2 and Lemma 3 we readily obtain $F(\mathcal{O}) \leq 8F\big(\text{OPT}(\cup_{i=1}^{m}\mathcal{S}_i)\big)$. Let AltGreedy() and DGDS() denote, respectively, Algorithm 2 and Algorithm 3 from the DGDS paper (Ghadiri & Schmidt, 2019). We can use Theorem 1 from Borodin et al. (Borodin et al., 2017) to obtain $F\big(\text{OPT}(\cup_{i=1}^{m}\mathcal{S}_i)\big) \leq 2F\big(\text{AltGreedy}(\cup_{i=1}^{m}\mathcal{S}_i)\big)$. This gives $F(\mathcal{O}) \leq 16F\big(\text{DGDS}(\mathcal{U})\big)$. $\blacksquare$

### B.5 PROOF OF LEMMA 5

*Proof.* Without loss of generality, suppose that $\text{ALG1}\big(\mathcal{C}|m, \lambda_c)\big)$ selected clusters $c_1, \ldots, c_m$. For each cluster $i$, let $\mathcal{U}_i \subseteq \mathcal{U}$ denote objects that belong to that cluster, and let $s_i^*$ denote an object with the highest quality score in that cluster, i.e., $s_i^* = \arg\max_{s \in \mathcal{U}_i} q(s)$.

Next, let $\mathcal{S}_i$ denote objects selected from that cluster by the algorithm, i.e., $\mathcal{S}_i = \text{ALG1}(\mathcal{U}_i|k, \lambda)$. It is clear that $s_i^* \in \mathcal{S}_i$. Also recall that we define quality score for the clusters as $q(c_i) = \text{median}(\{q(a) : a \in \mathcal{U}_i\})$. This gives $Q\big(\text{ALG1}(\mathcal{C})\big) \leq Q\big(\{s_1^*, \ldots, s_m^*\}\big)$.

Location of cluster $i$ is represented with cluster centroid $\mu_i$. We have $d(s_i^*, \mu_i) \leq r$ due to the definition of the radius $r$ as the distance from cluster centroid to the furthest point in the cluster. Therefore,

$$D\big(\text{ALG1}(\mathcal{C})\big) = D\big(\{\mu_1, \ldots, \mu_m\}\big) \leq D\big(\{s_1^*, \ldots, s_m^*\}\big) + rm(m-1). \tag{36}$$

We have that

$$F\big(\text{ALG}(\mathcal{C})|m\big) \leq F\big(\{s_1^*, \ldots, s_m^*\}|m\big) + rm(m-1). \tag{37}$$

Suppose $\{s_1^*, \ldots, s_m^*\} \subseteq \text{OPT}(\cup_{i=1}^m \mathcal{S}_i)$. Then, due to the nature of our objective function

$$F\big(\{s_1^*, \ldots, s_m^*\}|m\big) \leq F\big(\text{OPT}(\cup_{i=1}^m \mathcal{S}_i)|k\big). \tag{38}$$

Finally, suppose $\{s_1^*, \ldots, s_m^*\} \not\subseteq \text{OPT}(\cup_{i=1}^m \mathcal{S}_i)$, and $k \geq m$. Then if $F\big(\{s_1^*, \ldots, s_m^*\}|m\big) > F\big(\text{OPT}(\cup_{i=1}^m \mathcal{S}_i)|k\big)$, we could have replaced $m$ arbitrary points in $\text{OPT}(\cup_{i=1}^m \mathcal{S}_i)$ to get a higher value of $F(.|k)$. This would contradict the definition of $\text{OPT}(\cup_{i=1}^m \mathcal{S}_i)$ being the optimal set. Thus, again, we have

$$F\big(\{s_1^*, \ldots, s_m^*\}|m\big) \leq F\big(\text{OPT}(\cup_{i=1}^m \mathcal{S}_i)|k\big). \tag{39}$$

Combining this inequality with Eq. (37) gives the statement of the Lemma. ∎

### B.6 PROOF OF LEMMA 6

*Proof.* Our method clusters embeddings of objects in $\mathcal{U}$. Let $\mathcal{C}$ denote the set of clusters, and $\lambda_c$ denote the hyperparameter for cluster selection. We select clusters using $\text{ALG1}\big(\mathcal{C}|m, \lambda_c\big)$. We then select objects from each cluster, and finally select objects from the union of selections. We use $\lambda$ to denote the hyperparameter for objects selection.

Consider the union of points from selected clusters. The subset selected from this union that maximizes the objective is denoted as $\text{OPT}(\cup_{i=1}^m \mathcal{S}_i)$.

Next, consider any item $\boldsymbol{u} \in \mathcal{U}$, and let $\mu_{\boldsymbol{u}}$ denote the centroid of the cluster $\boldsymbol{u}$ belongs to. We introduce an auxiliary mapping $h_{\boldsymbol{u}}$ defined as follows. If the cluster of $\boldsymbol{u}$ is selected, $h_{\boldsymbol{u}}$ equals to $\mu_{\boldsymbol{u}}$. If the cluster of $\boldsymbol{u}$ is not selected, $h_{\boldsymbol{u}}$ equals to the nearest centroid among the selected clusters.

With these definitions in mind, and recalling that $d(.,.)$ is a metric, we have

$$D(\mathcal{O}) = \sum_{\boldsymbol{u} \in \mathcal{O}} \sum_{\boldsymbol{v} \in \mathcal{O}, \boldsymbol{v} \neq \boldsymbol{u}} d(\boldsymbol{u}, \boldsymbol{v}) \tag{40}$$

$$\leq \sum_{\boldsymbol{u} \in \mathcal{O}} \sum_{\boldsymbol{v} \in \mathcal{O}, \boldsymbol{v} \neq \boldsymbol{u}} \Big( d(\boldsymbol{u}, \mu_{\boldsymbol{u}}) + d(\mu_{\boldsymbol{u}}, h_{\boldsymbol{u}}) + d(h_{\boldsymbol{u}}, h_{\boldsymbol{v}}) + d(h_{\boldsymbol{v}}, \mu_{\boldsymbol{v}}) + d(\mu_{\boldsymbol{v}}, \boldsymbol{v}) \Big) \tag{41}$$

$$= 2(k-1) \sum_{\boldsymbol{z} \in \mathcal{O}} d(\boldsymbol{z}, \mu_{\boldsymbol{z}}) + 2(k-1) \sum_{\boldsymbol{z} \in \mathcal{O}} d(\mu_{\boldsymbol{z}}, h_{\boldsymbol{z}}) + \sum_{\boldsymbol{u} \in \mathcal{O}} \sum_{\boldsymbol{v} \in \mathcal{O}, \boldsymbol{v} \neq \boldsymbol{u}} d(h_{\boldsymbol{u}}, h_{\boldsymbol{v}}). \tag{42}$$

We now bound the three terms separately. Let $r$ denote the maximum radius among all clusters. We have that

$$2(k-1) \sum_{\boldsymbol{z} \in \mathcal{O}} d(\boldsymbol{z}, \mu_{\boldsymbol{z}}) \leq 2k(k-1)r. \tag{43}$$

Now consider the middle term. If the cluster of $\boldsymbol{z}$ is selected, $h_{\boldsymbol{z}}$ equals to $\mu_{\boldsymbol{z}}$ and $d(\boldsymbol{z}, \mu_{\boldsymbol{z}}) = 0$. If the cluster of $\boldsymbol{u}$ is not selected, Lemma 1 gives an upper bound. Therefore

$$2(k-1) \sum_{\boldsymbol{z} \in \mathcal{O}} d(\mu_{\boldsymbol{z}}, h_{\boldsymbol{u}}) \leq \frac{5k(k-1)}{(1-\lambda_c)m(m-1)} F(\text{ALG1}(\mathcal{C})) \tag{44}$$

$$\leq \frac{5k(k-1)}{(1-\lambda_c)m(m-1)} \Big( F(\text{OPT}(\cup_{i=1}^m \mathcal{S}_i)) + rm(m-1) \Big). \tag{45}$$

Finally, we bound the third term $\sum_{\boldsymbol{u} \in \mathcal{O}} \sum_{\boldsymbol{v} \in \mathcal{O}, \boldsymbol{v} \neq \boldsymbol{u}} d(h_{\boldsymbol{u}}, h_{\boldsymbol{v}})$.

Let $i = 1, \ldots, m$ index selected clusters in arbitrary order. Recall that $\mathcal{S}_i$ denotes objects selected from cluster $i$. Now consider an auxiliary set $\mathcal{S}_{\text{aux}}$, such that $|\mathcal{S}_{\text{aux}}| = k$, $\mathcal{S}_{\text{aux}} \subseteq \cup_{i=1}^m \mathcal{S}_i$, and $|\mathcal{S}_{\text{aux}} \cap \mathcal{S}_i| > 0$ for any $i$. In other words, $\mathcal{S}_{\text{aux}}$ contains at least one object from each selected cluster.

Due to the above definitions, for any $h_{\boldsymbol{u}}$ we know that (i) it is a centroid of a selected cluster, and (ii) we can find an object within that cluster that is included in $\mathcal{S}_{\text{aux}}$. Let $\boldsymbol{u}'$ and $\boldsymbol{v}'$ be such objects from clusters of $h_{\boldsymbol{u}}$ and $h_{\boldsymbol{v}}$, respectively.

We have that

$$\sum_{\boldsymbol{u} \in \mathcal{O}} \sum_{\boldsymbol{v} \in \mathcal{O}, \boldsymbol{v} \neq \boldsymbol{u}} d(h_{\boldsymbol{u}}, h_{\boldsymbol{v}}) \leq \sum_{\boldsymbol{u} \in \mathcal{O}} \sum_{\boldsymbol{v} \in \mathcal{O}, \boldsymbol{v} \neq \boldsymbol{u}} \big[ d(\boldsymbol{u}'(h_{\boldsymbol{u}}), \boldsymbol{v}'(h_{\boldsymbol{v}})) + 2r \big] \tag{46}$$

$$\leq 2rk(k-1) + \frac{1}{(1-\lambda)} F\Big( \mathcal{S}_{\text{aux}} | k \Big) \tag{47}$$

$$\leq 2rk(k-1) + \frac{1}{(1-\lambda)} F\Big( \text{OPT}(\cup_{i=1}^m \mathcal{S}_i) | k \Big). \tag{48}$$

Combining the three bounds gives

$$D(\mathcal{O}) < \Big( \frac{5k(k-1)}{(1-\lambda_c)m(m-1)} + \frac{1}{1-\lambda} \Big) F\big( \text{OPT}(\cup_{i=1}^m \mathcal{S}_i) \big) + \Big( \frac{5}{1-\lambda_c} + 4 \Big) rk(k-1). \tag{49}$$

∎

### B.7 Proof of Lemma 7

*Proof.* Let $\mathcal{S}^*$ denote the set of $k$ highest quality items from $\mathcal{U}$, i.e., $\mathcal{S}^* = \arg\max_{A \subseteq \mathcal{U}, |A|=k} \arg\max \sum_{\boldsymbol{u} \in A} q(\boldsymbol{u})$. Clearly, we can upper bound

$$Q(\mathcal{O}) \leq Q(\mathcal{S}^*) \leq \frac{1}{\lambda} F(\mathcal{S}^*) \tag{50}$$

$$\leq \frac{1}{\lambda} F\big( \text{OPT}(\mathcal{S}^*) \big) \tag{51}$$

$$\leq \frac{1}{\lambda} F\Big( \text{OPT}\big( \cup_{i=1}^m \mathcal{S}_i \cup \mathcal{S}^* \big) \Big). \tag{52}$$

∎

### B.8 Proof of Theorem 8

*Proof.* Using Lemma 7, we have $Q(\mathcal{O}) \leq \frac{1}{\lambda} F\Big( \text{OPT}\big( \cup_{i=1}^m \mathcal{S}_i \cup \mathcal{S}^* \big) \Big)$. Next, Lemma 6 gives $D(\mathcal{O}) \leq rk(k-1) \Big[ 4 + \frac{5}{1-\lambda_c} \Big] + F\Big( \text{OPT}(\cup_{i=1}^m \mathcal{S}_i) \Big) \Big[ \frac{5k(k-1)}{(1-\lambda_c)m(m-1)} + \frac{1}{(1-\lambda)} \Big]$.

Note that $F\Big( \text{OPT}(\cup_{i=1}^m \mathcal{S}_i) \Big) \leq F\Big( \text{OPT}\big( \cup_{i=1}^m \mathcal{S}_i \cup \mathcal{S}^* \big) \Big)$.

Let denote $\frac{\alpha}{2} \equiv 5 \frac{k(k-1)}{m(m-1)} \frac{(1-\lambda)}{(1-\lambda_c)} + 2$ and $\beta = k(k-1) \Big[ 4(1-\lambda) + 5 \frac{1-\lambda}{1-\lambda_c} \Big]$, we have that

$$F(\mathcal{O}) = \lambda Q(\mathcal{O}) + (1-\lambda) D(\mathcal{O}) \tag{53}$$

$$\leq \frac{\alpha}{2} F\Big( \text{OPT}\big( \cup_{i=1}^m \mathcal{S}_i \cup \mathcal{S}^* \big) \Big) + r\beta. \tag{54}$$

In other words,

$$F\Big(\text{OPT}\big(\cup_{i=1}^m \mathcal{S}_i \cup \mathcal{S}^*\big)\Big) \geq \frac{2}{\alpha} F(\mathcal{O}) - 2r\frac{\beta}{\alpha}. \tag{55}$$

According to Borodin et al. (Borodin et al., 2017), greedy selection where the quality term is scaled by $0.5$ is the half approximation of the optimal selection. We conclude that when $\sigma = 0.5$

$$F\big(\text{ALG2}_\sigma(\mathcal{U})\big) \geq \frac{1}{\alpha} F(\mathcal{O}) - r\frac{\beta}{\alpha}. \tag{56}$$

■

## C  ADDITIONAL EXPERIMENTAL DETAILS

### C.1  TECHNICAL DETAILS

The candidate retrieval task is performed using AWS instance *ml.r5.16xlarge* with $64$ CPUs, $10$ computational threads and $512$ GB RAM. For Figure 1, we also utilize another larger AWS instance *ml.r5.24xlarge* with $96$ CPUs, $25$ computational threads and $512$ GB RAM. The embedding dimension for candidate retrieval is $d = 1024$.

For both candidate retrieval and question answering tasks, MMR performance was evaluated on $\lambda$ values in $\{0.1, 0.3, 0.5, 0.7, 0.9\}$.

For question answering task, we used *us.anthropic.claude-3-5-haiku-20241022-v1:0*, with the idea that a smaller model complemented with RAG is a more cost-effective solution compared to using a much larger model. Also using a smaller model enabled us to see the effect of RAG more clearly. Next, prompt instructions included the following words:

```
You will be given a question and additional information to
    consider. This information might or might not be relevant to
    the question. Your task is to answer the question. Only use
    additional information if it's relevant.... (RAG results) ...
    (question) ... In your response, only include the answer
    itself. No tags, no other words.
```

For question and corpus embeddings, we used HuggingFaceEmbeddings.embed_documents() with default parameters. The embedding dimension is $d = 768$. Number of questions for each dataset was $50$.

In the results, MMR denotes greedy selection as per Algorithm 1. We have also evaluated greedy selection using the original maximum similarity criterion (Carbonell & Goldstein, 1998). Overall the results are slightly worse compared to the sum-based criterion, see Appendix Section C.6.

### C.2  ADDITIONAL INFORMATION ON CANDIDATE RETRIEVAL TASK

Our setting comes from the large-scale e-commerce platform where the real-time recommendation system (Deldjoo et al., 2024) includes two major steps: candidate retrieval (considered in this paper) and candidate ranking. The proposed MUSS has been deployed in real-world production for candidate retrieval, as part of the real-time recommendation, serving million customers daily.

We summarize the system in Figure 3. The first step: the candidate retrieval step returns $500$ products that are diverse and high quality. This candidate retrieval step is refreshed after every hour. The **quality score** is defined using an external ML model predicting the likelihood of an item being clicked on. This quality scores are precomputed offline and also refreshed after every hour. The entire corpus will be scored using this likelihood prediction.

The second step: the real-time ranking (less than 100ms) will be run on top of the above $500$ products to return a sorted list of $20$ products.

Table 6: Comparison on candidate retrieval to select $k = 500$ items. ✗ denotes that the algorithm did not complete within 12 hours of running. Our method achieves competitive performance and is faster than MMR and DGDS. Note that we focus on the Precision and Time as the main metrics for comparison while the other metrics are complementary. The highest precision score is in **bold**. The groundtruth for Amazon2M dataset is not available for evaluating Precision. Thus, it is used to compare running time.

Kitchen ($|\mathcal{U}| = 3872$, $\lambda = 0.9$)

| Method | $\lambda_c$ | Precision ↑ | Objective ↑ | Quality ↑ | Diversity ↑ | Time ↓ |
|---|---|---|---|---|---|---|
| random | | 50.0 | 0.687 | 0.693 | 0.638 | 0 |
| K-DPP | | 46.4 | 0.749 | 0.762 | 0.636 | 5.88 |
| clustering | | 61.6 | 0.879 | 0.906 | 0.641 | 0.59 |
| MMR | | 83.6 | 0.959 | 0.998 | 0.625 | 12.1 |
| DGDS | | 83.6 | 0.959 | 0.998 | 0.625 | 12.2 |
| MUSS(rand.A) | | 84.0 | 0.960 | 0.998 | 0.631 | 5.42 |
| MUSS(rand.B) | | 83.6 | 0.960 | 0.998 | 0.632 | 7.01 |
| MUSS | 0.1 | 95.5 | 0.954 | 0.998 | 0.644 | 6.34 |
| MUSS | 0.3 | 95.5 | 0.959 | 0.999 | 0.636 | 7.54 |
| MUSS | 0.5 | **95.7** | 0.959 | 0.999 | 0.633 | 8.11 |
| MUSS | 0.7 | **95.7** | 0.960 | 0.999 | 0.622 | 8.30 |
| MUSS | 0.9 | **95.7** | 0.960 | 0.999 | 0.618 | 8.24 |

Amazon100k ($|\mathcal{U}| = 108,258$, $\lambda = 0.9$)

| Method | $\lambda_c$ | Precision ↑ | Objective ↑ | Quality ↑ | Diversity ↑ | Time ↓ |
|---|---|---|---|---|---|---|
| random | | 11.2 | 0.730 | 0.736 | 0.674 | 0.0 |
| K-DPP | | ✗ | ✗ | ✗ | ✗ | ✗ |
| clustering | | 28.2 | 0.963 | 0.995 | 0.677 | 9.92 |
| MMR | | 39.4 | 0.970 | 0.999 | 0.711 | 311 |
| DGDS | | 39.4 | 0.970 | 0.999 | 0.711 | 271 |
| MUSS(rand.A) | | 42.8 | 0.969 | 0.999 | 0.698 | 49 |
| MUSS(rand.B) | | 41.6 | 0.969 | 0.999 | 0.700 | 53 |
| MUSS | 0.1 | 44.8 | 0.969 | 0.999 | 0.702 | 56 |
| MUSS | 0.3 | 42.8 | 0.970 | 0.999 | 0.705 | 54 |
| MUSS | 0.5 | 43.5 | 0.970 | 0.999 | 0.706 | 54 |
| MUSS | 0.7 | 44.4 | 0.970 | 0.999 | 0.704 | 53 |
| MUSS | 0.9 | **45.2** | 0.970 | 0.999 | 0.704 | 53 |

Amazon2M ($|\mathcal{U}| = 2M$, $\lambda = 0.9$)

| Method | $\lambda_c$ | Objective ↑ | Quality ↑ | Diversity ↑ | Time ↓ |
|---|---|---|---|---|---|
| random | | 0.659 | 0.515 | 0.659 | 0.0 |
| K-DPP | | ✗ | ✗ | ✗ | ✗ |
| clustering | | 0.666 | 0.983 | 0.666 | 17 |
| MMR | | **0.971** | 0.999 | 0.716 | 5870 |
| DGDS | | **0.971** | 0.999 | 0.716 | 114 |
| MUSS(rand.A) | | 0.970 | 0.999 | 0.710 | 72 |
| MUSS(rand.B) | | **0.971** | 0.999 | 0.716 | 73 |
| MUSS | 0.1 | 0.968 | 0.998 | 0.713 | 76 |
| MUSS | 0.3 | 0.969 | 0.998 | 0.715 | 74 |
| MUSS | 0.5 | **0.971** | 0.999 | 0.716 | 74 |
| MUSS | 0.7 | **0.971** | 0.999 | 0.716 | 73 |
| MUSS | 0.9 | **0.971** | 0.999 | 0.715 | 73 |

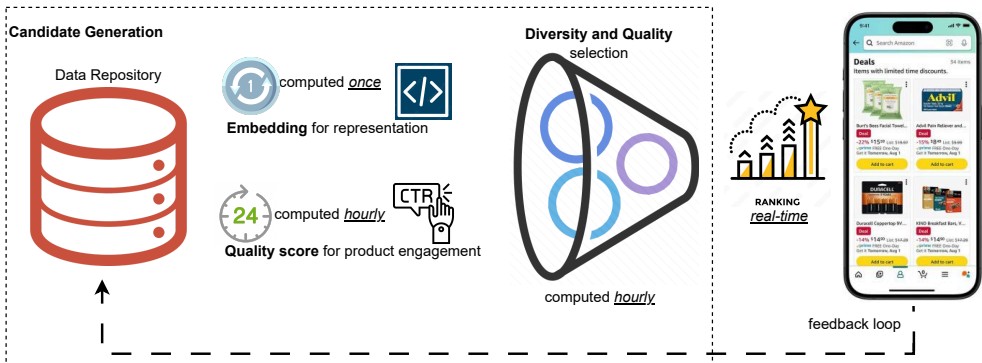

Figure 3: Flow chart of candidate retrieval module within the real-time ranking framework. The goal is to select the subset of $k$ products which are high quality and diverse every hour. We run this retrieval step per category and is non-personalized.

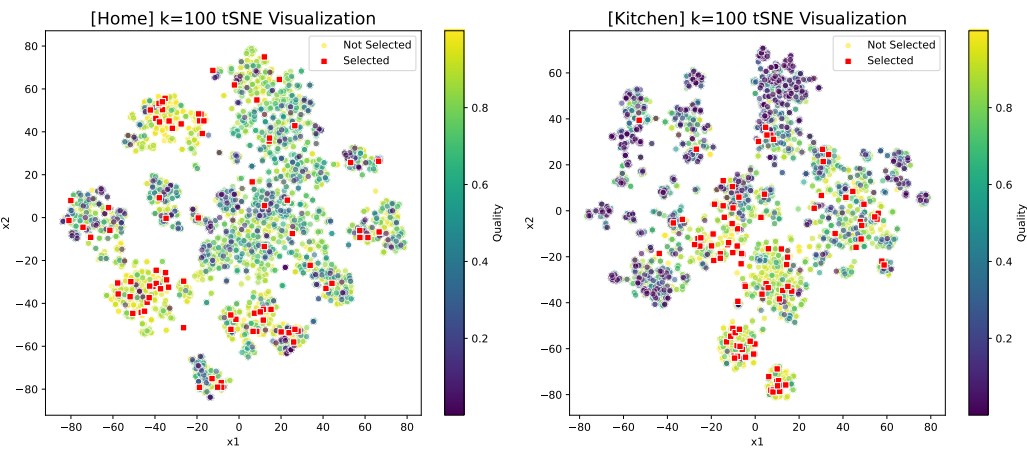

Figure 4: tSNE Visualization of selecting $k = 100$ items for "Home" and "Kitchen" datasets. Data forms clusters. Our method performs high-quality and diverse selection as shown by the red dots. The color scale indicates the quality score of the item.

Moreover, please note that items can typically be split into largely independent subsets (e.g., categories, such as books, baby food, etc.). Particularly, in our system, we retrieve 500 candidates per product category.

### C.3 ADDITIONAL RESULTS FOR CANDIDATE RETRIEVAL TASK

Full results for candidate item selection are presented in Table 6. The proposed MUSS consistently performs the best while significantly reduce the computational time. We note that while MMR will still find the highest objective function score since it directly maximizes Eq. (1), our MUSS also achieves comparable objective scores across four datasets.

Moreover, we have performed tSNE Visualization (Van der Maaten & Hinton, 2008) for selecting $k = 100$ items for "Home" and "Kitchen" datasets (Figure 4). We observe that the data forms coherent clusters. Our method tends to selects data points which are of high quality while being spread out within the space.

### C.4 VARYING $\lambda$ AND $\lambda_c$

In this study, we varied the trade-off parameters $\lambda_c$ (cluster-level selection) and $\lambda$ (item-level selection). We report the values of quality term $Q(\mathcal{S})$, diversity term $D(\mathcal{S})$, and the overall objective

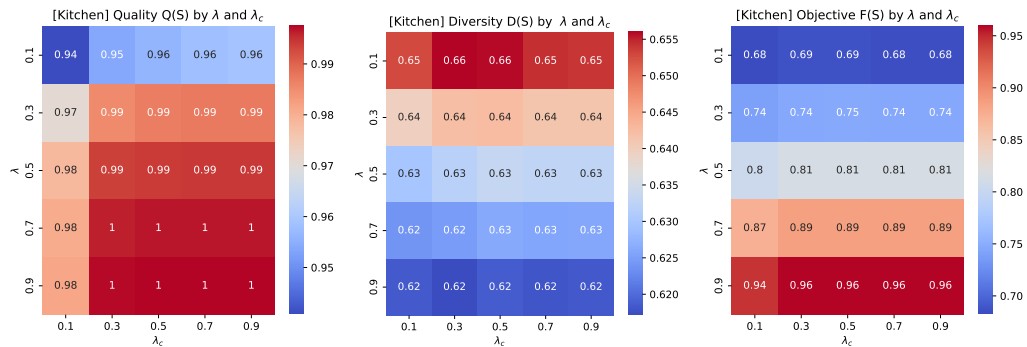

Figure 5: Diversity, quality, and the objective as the function of $\lambda_c$ and $\lambda$ for Kitchen dataset

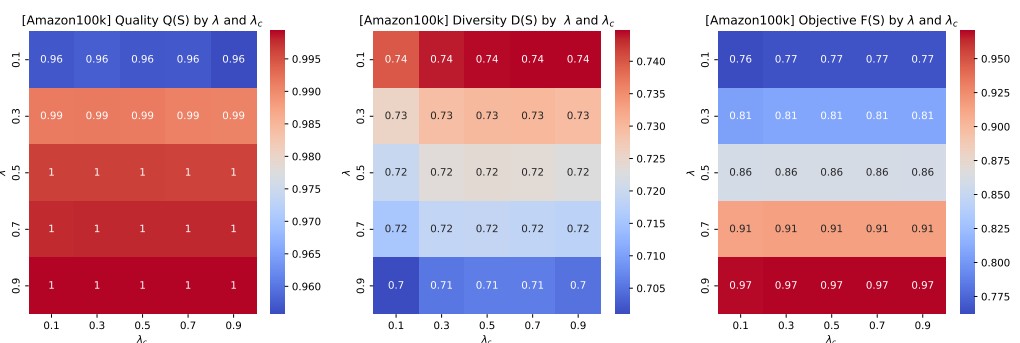

Figure 6: Diversity, quality, and the objective as the function of $\lambda_c$ and $\lambda$ for Amazon100k dataset

function $F(\mathcal{S})$ as defined in Eq. (1). Results are shown in Figures 5, and 6. As expected, when $\lambda$ increases, our objective function favours the quality term. Interestingly, for a fixed $\lambda$, the objective remains relatively stable at all values of $\lambda_c$.

### C.5 COMPUTATIONAL TIME FOR EACH COMPONENT IN DGDS AND MUSS

In Figure 7, we measure and report computational time spent in each component of Algorithm 2. This includes clustering (Line 1), greedy cluster selection (Line 3), greedy item selection in each selected cluster $\mathcal{S}$ (Line 5), and the final selection $\mathcal{S}$ (Line 7). In this setting, we select $k = 500$ items from Amazon2M datasets. We use different colors to indicate time spent in different steps. We consider two cases $k' = 50$ and $k' = 500$.

We can see that the running time is significantly faster when using $k' = 50$ (73 secs) against $k' = 500$ (510 secs), resulting in comparable objective function score of $0.971$ in Amazon2M dataset. Thus, it is preferable in practice to use a smaller value of $k' < k$.

While the DGDS does not spend time on clustering, it is slower than MUSS for two reasons: (i) there are more partitions ($l > m$) to be selecting from, and (ii) accordingly, after the union step $\cup_{i=1}^{l}\mathcal{S}_i$, the number of items is larger ($l \times k' > m \times k' + k$). In this setting, with the choices of $k = 500, l = 500, m = 100, k' = 50$, the number of items for DGDS ($25,000$) is significantly larger than MUSS ($5,500$) in the final selection. We note that point (i) can be potentially addressed for DGDS by using number of CPUs $p = l$. However, point (ii) remains a bottleneck for DGDS irrespective of getting more CPUs.

### C.6 COMPARING GREEDY OBJECTIVES

In our results, MMR denotes the sum-based greedy selection criterion as per Algorithm 1 ("sum-distance" criterion). We have also evaluated greedy selection using the original maximum similarity

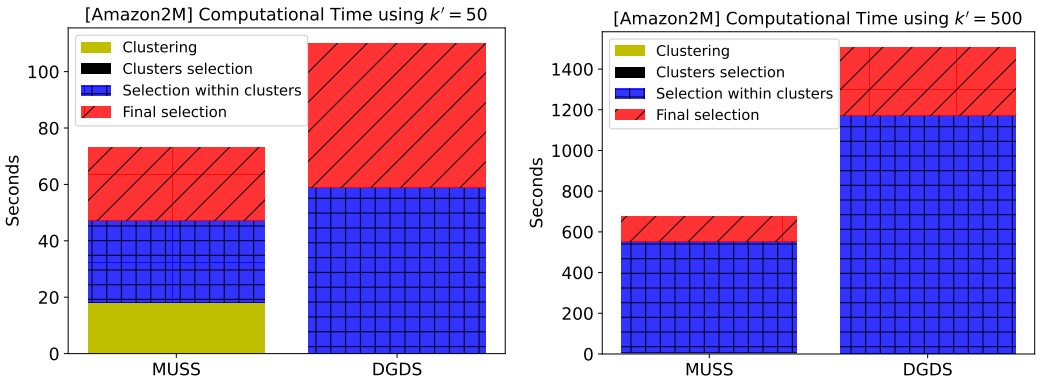

Figure 7: Computational time taken by each component of the Algorithm 2, compared against similar steps of DGDS. Our method is more computationally efficient due to having a smaller number of partitions and fewer data points in the final selection step (Line 7 Algorithm 2). Here, $k'$ is the number of data points selected within each cluster (Line 5 Algorithm 2). We note that if more number of CPUs $p = l$ is available for DGDS, then the time spent for selection within cluster (blue) will be similar for both DGDS and MUSS. However, the final selection (red) is still the bottleneck for DGDS.

criterion (Carbonell & Goldstein, 1998).

$$\text{MMR}'(\boldsymbol{s}) = \lambda \cdot \text{Sim}(\boldsymbol{s}, \boldsymbol{z}) - (1 - \lambda) \cdot \max_{t \in \mathcal{S}} \text{Sim}(\boldsymbol{s}, \boldsymbol{t}). \tag{57}$$

Here, $\boldsymbol{z}$ is the query for which MMR is performed, and $\mathcal{S}$ is the subset selected so far. For our quality and distance functions this criterion becomes

$$\text{MMR}'(\boldsymbol{s}) = \lambda \cdot q(\boldsymbol{s}) + (1 - \lambda) \cdot \min_{\boldsymbol{t} \in \mathcal{S}} d(\boldsymbol{s}, \boldsymbol{t}). \tag{58}$$

Overall the results were slightly worse compared to the "sum-distance" criterion, see Table 7.

### C.7 ABLATION OF MUSS WITHOUT TOP $k$ QUALITY ITEMS ADDITION

To facilitate the approximation bound analysis, the top $K$ highest quality items $\mathcal{S}^*$ have been added in Line 7 of Algorithm 2, $\mathcal{S} = \text{ALG1}(\cup_{i=1}^{m} \mathcal{S}_i \cup \mathcal{S}^*)$ for the refinement of the final selection.

We design an ablation study to test the empirical effect of this addition on Home and Amazon100k datasets. The comparison is presented in Table 8 using Precision as the key metric. Adding $\mathcal{S}^*$ in the final refinement results in a similar empirical performance. We propose to keep this addition as this step helps to tighten the Lemma 7.

Table 7: Precision achieved by MUSS using either "sum distance" or "min distance" as the greedy selection criterion.

| $\lambda_c$ | Diversity Distance | Precision ↑ | Objective ↑ | Quality ↑ | Diversity ↑ | Time ↓ |
|---|---|---|---|---|---|---|
| | Home ($|\mathcal{U}| = 4737, \lambda = 0.9$) | | | | | |
| 0.1 | sum distance | **74.5** | 0.962 | 0.996 | 0.643 | 7.12 |
| | min distance | 73.2 | 0.961 | 0.979 | 0.654 | 7.30 |
| 0.3 | sum distance | **74.2** | 0.962 | 0.997 | 0.646 | 7.86 |
| | min distance | 72.2 | 0.961 | 0.989 | 0.647 | 7.71 |
| 0.5 | sum distance | 74.0 | 0.962 | 0.997 | 0.646 | 8.91 |
| | min distance | 74.0 | 0.962 | 0.994 | 0.642 | 8.97 |
| 0.7 | sum distance | **74.1** | 0.962 | 0.997 | 0.647 | 9.17 |
| | min distance | 73.4 | 0.962 | 0.994 | 0.638 | 9.14 |
| 0.9 | sum distance | **74.8** | 0.962 | 0.997 | 0.648 | 8.18 |
| | min distance | 74.0 | 0.962 | 0.995 | 0.639 | 8.06 |
| | Amazon100K ($|\mathcal{U}| = 108,258, \lambda = 0.9$) | | | | | |
| 0.1 | sum distance | **44.8** | 0.970 | 0.999 | 0.703 | 56 |
| | min distance | 40.8 | 0.967 | 0.999 | 0.687 | 55 |
| 0.3 | sum distance | **42.8** | 0.970 | 0.999 | 0.705 | 55 |
| | min distance | 36.0 | 0.967 | 0.999 | 0.688 | 54 |
| 0.5 | sum distance | **43.5** | 0.970 | 0.999 | 0.706 | 55 |
| | min distance | 38.4 | 0.968 | 0.999 | 0.687 | 54 |
| 0.7 | sum distance | **44.4** | 0.970 | 0.999 | 0.706 | 53 |
| | min distance | 38.8 | 0.968 | 0.999 | 0.688 | 53 |
| 0.9 | sum distance | **45.2** | 0.970 | 0.999 | 0.705 | 53 |
| | min distance | 39.2 | 0.970 | 0.999 | 0.710 | 53 |

Table 8: Precision achieved by considering different versions of MUSS: in Line 7 of Algorithm 2 using either (i) $\mathcal{S} = \text{ALG1}\left(\cup_{i=1}^m \mathcal{S}_i \cup \mathcal{S}^* | k, \lambda\right)$ or (ii) $\mathcal{S} = \text{ALG1}\left(\cup_{i=1}^m \mathcal{S}_i \uplus \mathcal{S}^* | k, \lambda\right)$

| $\lambda_c$ | Diversity Distance | Precision ↑ | Objective ↑ | Quality ↑ | Diversity ↑ | Time ↓ |
|---|---|---|---|---|---|---|
| | Home ($|\mathcal{U}| = 4737, \lambda = 0.9$) | | | | | |
| 0.1 | $\mathcal{S} = \text{ALG1}(\cup_{i=1}^m \mathcal{S}_i \cup \mathcal{S}^*)$ | **74.5** | 0.962 | 0.996 | 0.643 | 7.12 |
| | $\mathcal{S} = \text{ALG1}(\cup_{i=1}^m \mathcal{S}_i \uplus \mathcal{S}^*)$ | 72.7 | 0.961 | 0.979 | 0.654 | 7.30 |
| 0.3 | $\mathcal{S} = \text{ALG1}(\cup_{i=1}^m \mathcal{S}_i \cup \mathcal{S}^*)$ | **74.2** | 0.962 | 0.997 | 0.646 | 7.86 |
| | $\mathcal{S} = \text{ALG1}(\cup_{i=1}^m \mathcal{S}_i \uplus \mathcal{S}^*)$ | 74.2 | 0.962 | 0.989 | 0.647 | 7.71 |
| 0.5 | $\mathcal{S} = \text{ALG1}(\cup_{i=1}^m \mathcal{S}_i \cup \mathcal{S}^*)$ | **74.0** | 0.962 | 0.997 | 0.646 | 8.91 |
| | $\mathcal{S} = \text{ALG1}(\cup_{i=1}^m \mathcal{S}_i \uplus \mathcal{S}^*)$ | 73.7 | 0.961 | 0.994 | 0.642 | 8.97 |
| 0.7 | $\mathcal{S} = \text{ALG1}(\cup_{i=1}^m \mathcal{S}_i \cup \mathcal{S}^*)$ | 74.1 | 0.962 | 0.997 | 0.647 | 9.17 |
| | $\mathcal{S} = \text{ALG1}(\cup_{i=1}^m \mathcal{S}_i \uplus \mathcal{S}^*)$ | **75.2** | 0.962 | 0.994 | 0.638 | 9.14 |
| 0.9 | $\mathcal{S} = \text{ALG1}(\cup_{i=1}^m \mathcal{S}_i \cup \mathcal{S}^*)$ | 74.8 | 0.962 | 0.997 | 0.648 | 8.18 |
| | $\mathcal{S} = \text{ALG1}(\cup_{i=1}^m \mathcal{S}_i \uplus \mathcal{S}^*)$ | 74.8 | 0.962 | 0.995 | 0.639 | 8.06 |

| $\lambda_c$ | Setting | Precision ↑ | Objective ↑ | Quality ↑ | Diversity ↑ | Time ↓ |
|---|---|---|---|---|---|---|
| | Amazon100K ($|\mathcal{U}| = 108,258, \lambda = 0.9$) | | | | | |
| 0.1 | $\mathcal{S} = \text{ALG1}(\cup_{i=1}^m \mathcal{S}_i \cup \mathcal{S}^*)$ | **44.8** | 0.970 | 0.999 | 0.703 | 56 |
| | $\mathcal{S} = \text{ALG1}(\cup_{i=1}^m \mathcal{S}_i \uplus \mathcal{S}^*)$ | 40.4 | 0.967 | 0.999 | 0.686 | 54 |
| 0.3 | $\mathcal{S} = \text{ALG1}(\cup_{i=1}^m \mathcal{S}_i \cup \mathcal{S}^*)$ | 42.8 | 0.970 | 0.999 | 0.705 | 55 |
| | $\mathcal{S} = \text{ALG1}(\cup_{i=1}^m \mathcal{S}_i \uplus \mathcal{S}^*)$ | 42.8 | 0.967 | 0.999 | 0.694 | 55 |
| 0.5 | $\mathcal{S} = \text{ALG1}(\cup_{i=1}^m \mathcal{S}_i \cup \mathcal{S}^*)$ | 43.5 | 0.970 | 0.999 | 0.706 | 55 |
| | $\mathcal{S} = \text{ALG1}(\cup_{i=1}^m \mathcal{S}_i \uplus \mathcal{S}^*)$ | **44.6** | 0.969 | 0.999 | 0.693 | 56 |
| 0.7 | $\mathcal{S} = \text{ALG1}(\cup_{i=1}^m \mathcal{S}_i \cup \mathcal{S}^*)$ | 44.4 | 0.970 | 0.999 | 0.706 | 53 |
| | $\mathcal{S} = \text{ALG1}(\cup_{i=1}^m \mathcal{S}_i \uplus \mathcal{S}^*)$ | **45.8** | 0.968 | 0.999 | 0.685 | 54 |
| 0.9 | $\mathcal{S} = \text{ALG1}(\cup_{i=1}^m \mathcal{S}_i \cup \mathcal{S}^*)$ | **45.2** | 0.970 | 0.999 | 0.705 | 53 |
| | $\mathcal{S} = \text{ALG1}(\cup_{i=1}^m \mathcal{S}_i \uplus \mathcal{S}^*)$ | 45.0 | 0.968 | 0.999 | 0.686 | 56 |

