# OpenReview forum: "MUSS: Multilevel Subset Selection for Relevance and Diversity"
_ICLR.cc/2026/Conference — ICLR 2026 Conference Withdrawn Submission_

### Official Review · Reviewer_AtAh · 2025-10-24

**Soundness:** 3
**Presentation:** 2
**Contribution:** 2
**Rating:** 2
**Confidence:** 3

**Summary:**

The paper introduces MUSS (Multilevel Subset Selection), a scalable algorithm for selecting relevant and diverse subsets from large datasets. MUSS extends greedy and distributed approaches (e.g., DGDS, MMR) through a clustering framework that improves efficiency and theoretical guarantees. The authors claim a tighter approximation bound (1/16 vs. 1/31) and substantial runtime improvements, with evaluations on recommender and RAG tasks showing consistent but modest gains.

**Strengths:**

1. The paper is well-written and easy to follow. The motivation for balancing relevance and diversity is clear, and the introduction effectively contextualizes the problem within ML applications such as recommendation and RAG.

2. A notable contribution is the new theoretical analysis framework that connects cluster-level and item-level objectives (Lemma 5) and yields a tighter approximation bound for DGDS. While not groundbreaking, this provides useful analytical insight into distributed subset selection problems.

**Weaknesses:**

1. Despite claims of scalability, the experimental datasets are relatively small and few. Only four datasets are used, and many are from internal sources, limiting reproducibility. Moreover, the largest dataset (“Amazon2M”) lacks ground truth labels for precision evaluation, weakening the empirical validation of effectiveness at scale.
2. The study lacks a baseline evaluating relevance-only selection, i.e., choosing top-k items by quality without diversity constraints. This would clarify how much of the gain is due to MUSS’s balancing strategy versus its relevance modeling.
3. While MUSS introduces a multilevel structure, the underlying approach — greedy subset selection with clustering — resembles prior works e.g. [1], [2] and [3]. As such, the paper’s novelty feels incremental. The theoretical tightening and practical scaling improvements are valuable but do not fully reposition the method in a distinct conceptual space.
4. While the theoretical results are mathematically sound, they do not directly translate to clear practical insights. For example, the constants and bounds are not empirically verified, and their impact on performance is unclear.

[1] Uncovering the Bigger Picture: Comprehensive Event Understanding via Diverse News Retrieval

[2] Diversity-Aware k-Maximum Inner Product Search Revisited

[3] Solving Diversity-Aware Maximum Inner Product Search Efficiently and Effectively

**Questions:**

1. Could the authors provide additional large-scale or publicly available datasets to better substantiate MUSS’s claimed scalability and reproducibility, especially given that the largest dataset (“Amazon2M”) lacks ground truth labels?
2. Would the authors include a relevance-only (top-k by quality) baseline to clarify how much of the observed improvement comes from MUSS’s diversity mechanism versus its relevance modeling?
3. How does MUSS fundamentally differ from prior clustering-based or diversity-aware retrieval methods?
4. Can the authors empirically validate the proposed theoretical bounds to demonstrate their practical significance and impact on real-world performance?

---

### Official Review · Reviewer_9FtG · 2025-10-29

**Soundness:** 3
**Presentation:** 2
**Contribution:** 2
**Rating:** 4
**Confidence:** 4

**Summary:**

The paper presents a new algorithm for the problem of selecting k items, from a large dataset, so as to maximize a function that combines relevance and pairwise distance between the selected items. The problem is motivated by applications that require diversity-aware item selection. The specific challenge is to solve this problem in a distributed setting in order to make the method scalable to large datasets. The presented algorithm leverages the cluster structure of the data (i.e., it clusters the data and operates on the clusters) and offers improved approximation guarantee compared to previous state-of-the-art distributed algorithm.

**Strengths:**

S1. The problem studied in the paper is well motivated and relevant to the ICLR community.
S2. Improved theoretical result over the state-of-the-art method.
S3. Interesting idea of clustering the data and eliminating whole clusters. It is also interesting how the items are selected from the clusters as candidates for the selection in the last phase.
S4. Empirical results show modest improvement over baselines.

**Weaknesses:**

W1. While I found the proposed algorithm interesting, I think that the overall contribution is somewhat incremental. First, while there are technical differences from the previous distributed algorithm, the overall idea and the setting is quite similar. Second, the theoretical improvement is not so impressive. Third, the standard greedy algorithm can also be parallelized (by parallelizing the greedy selection step) and it overs a 1/2 approximation guarantee (by the max-sum diversification result of Borodin et al.). Given this, the improvement of the standard greedy is fairly marginal.
W2. Similarly to the theoretically marginal improvement, the empirical evaluation also shows marginal improvement over the baselines. In particular, the speedup is not impressive compared to the standard greedy.

**Questions:**

Please address W1 and W2 above.

---

### Official Review · Reviewer_nNP5 · 2025-10-31

**Soundness:** 4
**Presentation:** 3
**Contribution:** 3
**Rating:** 8
**Confidence:** 2

**Summary:**

This paper addresses the problem of selecting a relevant and diverse subset of items, which is critical in applications like recommender systems and retrieval-augmented generation (RAG). The authors propose MUSS (Multilevel Subset Selection), a scalable and efficient algorithm that combines clustering-based pruning with a multi-stage greedy selection mechanism. Unlike prior methods such as MMR and DGDS, MUSS exploits data structure to reduce computational complexity and improve performance. Theoretical guarantees show that MUSS achieves a constant-factor approximation of the optimal objective, and even tightens the known bounds for DGDS. Empirical results show that MUSS outperforms baselines on both precision and runtime across various datasets, and it has been successfully deployed in a real-world production system.

**Strengths:**

* This paper proposes a multilevel selection strategy for relevant and diverse subset selection, supported by a constant-factor approximation guarantee derived from rigorous theoretical analysis.
* The method shows scalability, achieving up to 80× speedup over MMR and notable gains over DGDS, while maintaining or improving selection quality. It has also been deployed in a real-world production system serving millions of users.
* The use of clustering to prune candidate sets before final selection is novel and effective, offering both computational savings and improved selection by leveraging latent structure in the data.

**Weaknesses:**

* The paper introduces several hyperparameters (e.g., $\lambda$, $\lambda_c$, number of clusters), but provides limited practical guidance on how to tune them. This may hinder reproducibility or ease of adoption in real-world systems.
* The evaluation on RAG tasks is relatively narrow, using only two datasets and 50 questions, which may not be sufficient to claim broader generalization.
* Some components of the algorithm, such as the inclusion of top-k quality items in the final selection step, appear to have only marginal impact in practice, raising questions about their necessity.

**Questions:**

* Have you evaluated how different clustering algorithms affect MUSS performance? The method's reliance on clustering suggests it could be sensitive to this choice.
* In production or practical settings, how are the hyperparameters ($\lambda$, $\lambda_c$, number of clusters l, m) chosen? Are fixed values used, or is there a tuning strategy or heuristic?
* Given that the top-k quality item addition step leads to less than 0.3 percentage point change in performance on average, could this step be safely omitted in practice?

---

### Official Review · Reviewer_eGut · 2025-11-01

**Soundness:** 3
**Presentation:** 2
**Contribution:** 2
**Rating:** 2
**Confidence:** 3

**Summary:**

This paper proposes a multi-level subset selection method that balances relevance and diversity. The approach consists of three steps: (1) applying k-means clustering to partition the data, (2) performing greedy selection at the cluster level, and (3) choosing the final subset by selecting the top-k items with the highest quality scores.

**Strengths:**

+  Considering both quality and diversity in subset selection is important for recommendation systems in certain applications.

+ The paper provides a theoretical analysis showing that the proposed method achieves a constant-factor approximation.

+  The method is shown to be deployed in a real-world application, highlighting its practical impact.

**Weaknesses:**

- As a scientific research paper, the novelty appears limited. The main idea of the paper seems to be separating the selection process into a clustering stage (Step 1) and a greedy selection stage (Step 2). Both the use of k-means in Step 1 and the greedy method in Step 2 are textbook-level techniques.

- The balance between quality and diversity is achieved using a simple linear weighting scheme, with weights manually defined. The paper does not justify this weighting choice. Is the contribution of quality and diversity to subset selection truly linear?

- The paper primarily compares against DGDS (2019). However, numerous recent learning-based methods leveraging modern neural architectures have been introduced since then. State-of-the-art learning-based methods are not analyzed or included in the experimental comparison.

-  Following from the above point, the baselines used in the experiments are relatively outdated for the research community, making the claimed research novelty of the proposed method unconvincing.

- The paper does not clearly explain how the proposed method is executed in a distributed setting. Is the distributed mechanism identical to DGDS, with the only modification occurring in the final selection step (Step 3)?

**Questions:**

See the weakness section.

---

### Note · Authors · 2025-11-22

I have read and agree with the venue's withdrawal policy on behalf of myself and my co-authors.